# S-NeRF: Neural Radiance Fields for Street Views

**Ziyang Xie**[1]\*, **Junge Zhang**[1]\*, **Wenye Li**[1], **Feihu Zhang**[2], **Li Zhang**[1]†
[1]Fudan University  [2]University of Oxford
https://ziyang-xie.github.io/s-nerf

## ABSTRACT

Neural Radiance Fields (NeRFs) aim to synthesize novel views of objects and scenes, given the object-centric camera views with large overlaps. However, we conjugate that this paradigm does not fit the nature of the street views that are collected by many self-driving cars from the large-scale unbounded scenes. Also, the onboard cameras perceive scenes without much overlapping. Thus, existing NeRFs often produce blurs, "floaters" and other artifacts on street-view synthesis. In this paper, we propose a new *street-view NeRF* (*S-NeRF*) that considers novel view synthesis of both the large-scale background scenes and the foreground moving vehicles jointly. Specifically, we improve the scene parameterization function and the camera poses for learning better neural representations from street views. We also use the the noisy and sparse LiDAR points to boost the training and learn a robust geometry and reprojection based confidence to address the depth outliers. Moreover, we extend our S-NeRF for reconstructing moving vehicles that is impracticable for conventional NeRFs. Thorough experiments on the large-scale driving datasets (*e.g.*, nuScenes and Waymo) demonstrate that our method beats the state-of-the-art rivals by reducing $7 \sim 40\%$ of the mean-squared error in the street-view synthesis and a 45% PSNR gain for the moving vehicles rendering.

## 1 INTRODUCTION

Neural Radiance Fields (Mildenhall et al., 2020) have shown impressive performance on photorealistic novel view rendering. However, original NeRF is usually designed for object-centric scenes and require camera views to be heavily overlapped (as shown in Figure 1(a)).

Recently, more and more street view data are collected by self-driving cars. The reconstruction and novel view rendering for street views can be very useful in driving simulation, data generation, AR and VR. However, these data are often collected in the unbounded outdoor scenes (*e.g.* nuScenes (Caesar et al., 2019) and Waymo (Sun et al., 2020) datasets). The camera placements of such data acquisition systems are usually in a panoramic settings without object-centric camera views (Figure 1(b)). Moreover, the overlaps between adjacent camera views are too small to be effective for training NeRFs. Since the ego car is moving fast, some objects or contents only appear in a limited number of image views. (*e.g.* Most of the vehicles need to be reconstructed from just $2 \sim 6$ views.) All these problems make it difficult to optimize existing NeRFs for street-view synthesis.

MipNeRF-360 (Barron et al., 2022) is designed for training in unbounded scenes. BlockNeRF (Tancik et al., 2022) proposes a block-combination strategy with refined poses, appearances, and exposure on the MipNeRF (Barron et al., 2021) base model for processing large-scale outdoor scenes. However, they still require enough intersected camera rays (Figure 1(a)) and large overlaps across different cameras (*e.g.* Block-NeRF uses a special system with twelve cameras for data acquisition to guarantee enough overlaps between different camera views). They produce many blurs, "floaters" and other artifacts when training on existing self-driving datasets (*e.g.* nuScenes (Caesar et al., 2019) and Waymo (Sun et al., 2020), as shown in Figure 2(a).

Urban-NeRF (Rematas et al., 2022) takes accurate dense LiDAR depth as supervision for the reconstruction of urban scenes. However, these dense LiDAR depths are difficult and expensive to collect.

---

\*Equal contribution
†Corresponding author with School of Data Science, Fudan University (lizhangfd@fudan.edu.cn).

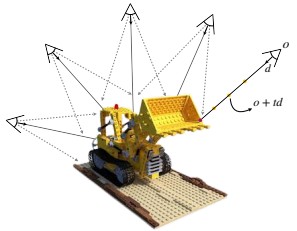
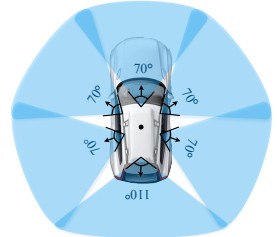

(a) NeRFs camera settings         (b) Ego car camera settings

Figure 1: Problem illustration. (a) Conventional NeRFs Mildenhall et al. (2020); Barron et al. (2021) require object-centric camera views with large overlaps. (b) In the challenging large-scale outdoor driving scenes Caesar et al. (2019); Sun et al. (2020)), the camera placements for data collection are usually in a panoramic view settings. Rays from different cameras barely intersect with others in the unbounded scenes. The overlapped field of view between adjacent cameras is too small to be effective for training the existing NeRF models.

As shown in Figure 3(a), data Caesar et al. (2019); Sun et al. (2020) collected by self-driving cars can not be used for training Urban-NeRF because they only acquire sparse LiDAR points with plenty of outliers when projected to images (*e.g.* only 2∼5K points are captured for each nuScenes image).

In this paper, we contribute a new NeRF design (S-NeRF) for the novel view synthesis of both the large-scale (background) scenes and the foreground moving vehicles. Different from other large-scale NeRFs (Tancik et al., 2022; Rematas et al., 2022), our method does not require specially designed data acquisition platform used in them. Our S-NeRF can be trained on the standard self-driving datasets (*e.g.* nuScenes (Caesar et al., 2019) and Waymo (Sun et al., 2020)) that are collected by common self-driving cars with fewer cameras and noisy sparse LiDAR points to synthesize novel street views.

We improve the scene parameterization function and the camera poses for learning better neural representations from street views. We also develop a novel depth rendering and supervision method using the noisy sparse LiDAR signals to effectively train our S-NeRF for street-view synthesis. To deal with the depth outliers, we propose a new confidence metric learned from the robust geometry and reprojection consistencies.

Not only for the background scenes, we further extend our S-NeRF for high-quality reconstruction of the moving vehicles (*e.g.* moving cars) using the proposed virtual camera transformation.

In the experiments, we demonstrate the performance of our S-NeRF on the standard driving datasets (Caesar et al., 2019; Sun et al., 2020). For the static scene reconstruction, our S-NeRF far outperforms the large-scale NeRFs (Barron et al., 2021; 2022; Rematas et al., 2022). It reduces the mean-squared error by $7 \sim 40\%$ and produces impressive depth renderings (Figure 2(b)). For the foreground objects, S-NeRF is shown capable of reconstructing moving vehicles in high quality, which is impracticable for conventional NeRFs (Mildenhall et al., 2020; Barron et al., 2021; Deng et al., 2022). It also beats the latest mesh-based reconstruction method Chen et al. (2021b), improving the PSNR by 45% and the structure similarity by 18%.

## 2 RELATED WORK

### 2.1 3D RECONSTRUCTION

Traditional reconstruction and novel view rendering (Agarwal et al., 2011) often rely on Structure-from-Motion (SfM), multi-view stereo and graphic rendering (Losasso & Hoppe, 2004).

Learning-based approaches have been widely used in 3D scene and object reconstruction (Sitzmann et al., 2019; Xu et al., 2019; Engelmann et al., 2021). They encode the feature through a deep neural network and learn various geometry representations, such as voxels (Kar et al., 2017; Sitzmann et al., 2019), patches (Groueix et al., 2018) and meshes (Wang et al., 2018; Chen et al., 2021b).

### 2.2 NEURAL RADIANCE FIELDS

Neural Radiance Fields (NeRF) is proposed in (Mildenhall et al., 2020) as an implicit neural representation for novel view synthesis. Various types of NeRFs haven been proposed for acceleration. (Yu et al., 2021a; Rebain et al., 2021), better generalization abilities (Yu et al., 2021b; Trevithick &

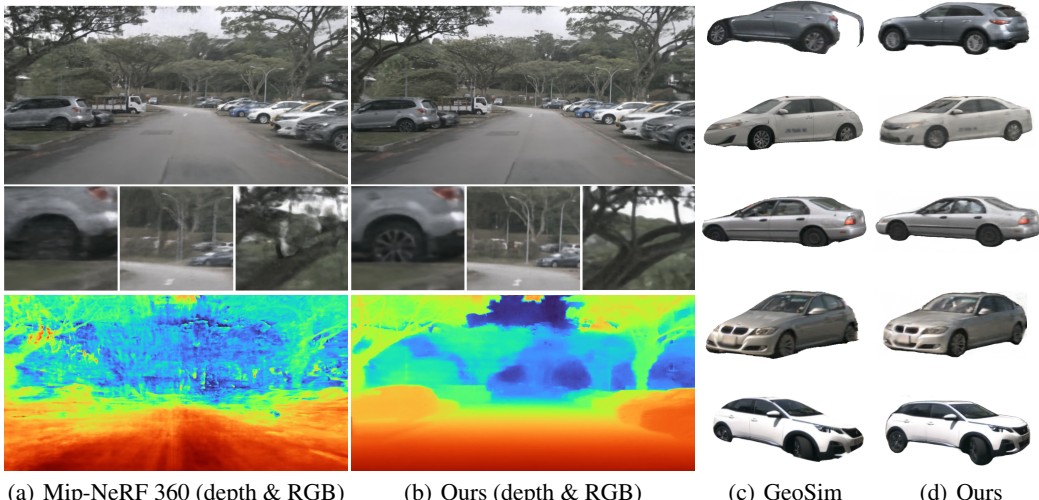

|                             |                        |            |        |
| :-------------------------: | :--------------------: | :--------: | :----: |
| (a) Mip-NeRF 360 (depth & RGB) | (b) Ours (depth & RGB) | (c) GeoSim | (d) Ours |

Figure 2: Performance illustration in novel view rendering on a challenging nuScenes scene Caesar et al. (2019), (a) the state-of-the-art method Barron et al. (2022) produces poor results with blurred texture details and plenty of depth errors, (b) our S-NeRF can achieve accurate depth maps and fine texture details with fewer artifacts. (d) Our method can also be used for the reconstruction of moving vehicles which is impossible for previous NeRFs. It can synthesize better novel views compared with the mesh method Chen et al. (2021b).

Yang, 2021), new implicit modeling functions (Yariv et al., 2021; Wang et al., 2021a), large-scale scenes (Tancik et al., 2022; Zhang et al., 2020), and depth supervised training (Deng et al., 2022; Rematas et al., 2022).

**Large-scale NeRF**  Many NeRFs have been proposed to address the challenges of large-scale outdoor scenes. NeRF in the wild (Martin-Brualla et al., 2021) applies appearance and transient embeddings to solve the lighting changes and transient occlusions. Neural plenoptic sampling (Li et al., 2021b) proposes to use a Multi-Layer Perceptron (MLP) as an approximator to learn the plenoptic function and represent the light-field in NeRF. Mip-NeRF (Barron et al., 2021) develops a conical frustum encoding to better encode the scenes at a continuously-valued scale. Using Mip-NeRF as a base block, Block-NeRF (Tancik et al., 2022) employs a block-combination strategy along with pose refinement, appearance, and exposure embedding on large-scale scenes. Mip-NeRF 360 (Barron et al., 2022) improves the Mip-NeRF for unbounded scenes by contracting the whole space into a bounded area to get a more representative position encoding.

**Depth supervised NeRF**  DS-NeRF (Deng et al., 2022) utilizes the sparse depth generated by COLMAP (Schönberger & Frahm, 2016) to supervise the NeRF training. PointNeRF (Xu et al., 2022) uses point clouds to boost the training and rendering with geometry constraints. DoN-eRF (Neff et al., 2021) realizes the ray sampling in a log scale and uses the depth priors to improve ray sampling. NerfingMVS (Wei et al., 2021) also instructs the ray sampling during training via depth and confidence. Dense depth priors are used in (Roessle et al., 2022) which are recovered from sparse depths.

These methods, however, are designed for processing small-scale objects or indoor scenes. Urban-NeRF (Rematas et al., 2022) uses accurate dense LiDAR depth as supervision to learn better reconstruction of the large-scale urban scenes. But these dense LiDAR depths are difficult and expensive to collect. The common data collected by self-driving cars (Caesar et al., 2019; Sun et al., 2020) can not be used in training Urban-NeRF because they only acquire noisy and sparse LiDAR points. In contrast, our S-NeRF can use such defect depths along with a learnable confidence measurement to learn better neural representations for the large-scale street-view synthesis.

## 3    NeRF for Street Views

In this section, we present our S-NeRF that can synthesize photo-realistic novel-views for both the large-scale background scenes and the foreground moving vehicles (Section 3.3).

| (a) Noisy sparse points | (b) Our depth supervision | (c) Learned confidence | (d) Our depth rendering |

Figure 3: Depth supervision and rendering.

In the street views, there are many dynamic objects. In order to be used for self-driving simulation or VR applications, dynamic objects must be fully controllable and move in controlled locations, speed, and trajectories. Therefore, the background scenes and the foreground vehicles must be reconstructed separately and independently.

We propose a novel NeRF design that uses sparse noisy LiDAR signals to boost the robust reconstruction and novel street-view rendering. Pose refinement and the virtual camera transform are added to achieve accurate camera poses (Section 3.2). To deal with the outliers in the sparse LiDAR depths, we use a depth completion network Park et al. (2020) to propagate the sparse depth and employ a novel confidence measurement based on the robust reprojection and geometry confidence (Section 3.4). Finally, S-NeRF is trained using the proposed depth and RGB losses (Section 3.5).

## 3.1 PRELIMINARY

Neural Radiance Field (NeRF) represents a scene as a continuous radiance field and learns a mapping function $f : (\mathbf{x}, \theta) \to (\mathbf{c}, \sigma)$. It takes the 3D position $\mathrm{x}_i \in \mathbb{R}^3$ and the viewing direction $\theta_i$ as input and outputs the corresponding color $c_i$ with its differential density $\sigma_i$. The mapping function is realized by two successive multi-layer perceptrons (MLPs).

NeRF uses the volume rendering to render image pixels. For each 3D point in the space, its color can be rendered through the camera ray $\mathbf{r}(t) = \mathbf{o} + t\mathbf{d}$ with $N$ stratified sampled bins between the near and far bounds of the distance. The output color is rendered as:

$$\hat{\mathbf{I}}(\mathbf{r}) = \sum_{i=1}^{N} T_i(1 - e^{-\sigma_i \delta_i})\mathbf{c}_i, \quad T_i = \exp\left(-\sum_{j=1}^{i-1} \sigma_j \delta_j\right) \tag{1}$$

where $\mathbf{o}$ is the origin of the ray, $T_i$ is the accumulated transmittance along the ray, $\mathbf{c}_i$ and $\sigma_i$ are the corresponding color and density at the sampled point $t_i$. $\delta_j = t_{j+1} - t_j$ refers to the distance between the adjacent point samples.

## 3.2 CAMERA POSE PROCESSING

SfM (Schönberger & Frahm, 2016) used in previous NeRFs fails in computing camera poses for the self-driving data since the camera views have fewer overlaps (Figure 1(b)). Therefore, we proposed two different methods to reconstruct the camera poses for the static background and the foreground moving vheicles.

**Background scenes** For the static background, we use the camera parameters achieved by sensor-fusion SLAM and IMU of the self-driving cars (Caesar et al., 2019; Sun et al., 2020) and further reduce the inconsistency between multi-cameras with a learning-based pose refinement network. We follow Wang *et al.* Wang et al. (2021b) to implicitly learn a pair of refinement offsets $\Delta P = (\Delta R, \Delta T)$ for each original camera pose $P = [R, T]$, where $R \in SO(3)$ and $T \in \mathbb{R}^3$. The pose refine network can help us to ameliorate the error introduced in the SLAM algorithm and make our system more robust.

**Moving vehicles** While the method proposed above is appropriate for the static background, camera pose estimation of moving objects is especially difficult due to the complicated movements of both the ego car and the target objects. As illustrated in Figure 4, we compute the relative position $\hat{P}$ between the camera and the target object. We use the center of the target object as the origin of the coordinate system. In the experiments, we use the 3D detectors (*e.g.* Yin et al. (2021)) to detect the 3D bounding box and the center of the target object.

Using the ego car's center as the coordinate system's origin. $P_b$ represents the position of the target object. $P_i$ is the position of the camera $i$ (there are 5∼6 cameras on the ego car). We now transform

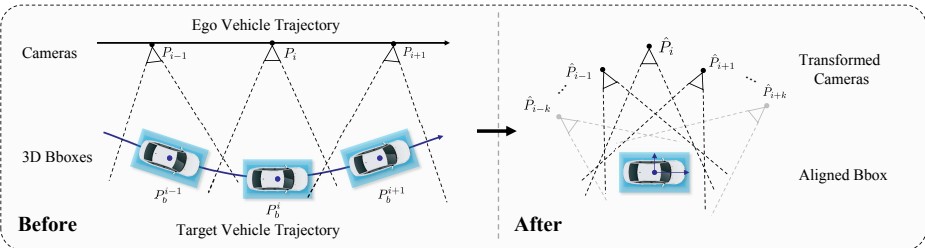

Figure 4: Illustration of our camera transformation process for moving vehicles. During the data collection, the ego car (camera) is moving and the target car (object) is also moving. The virtual camera system treats the target car (moving object) as static and then compute the relative camera poses for the ego car's camera. These relative camera poses can be estimated through the 3D object detectors. After the transformation, only the camera is moving which is favorable in training NeRFs.

the coordinate system by setting the target object's center as the coordinate system's origin (as illustrated in Figure 4).

$$\hat{P}_i = (P_i P_b^{-1})^{-1} = P_b P_i^{-1}, \quad P^{-1} = \begin{bmatrix} R^T & -R^T T \\ \mathbf{0}^T & 1 \end{bmatrix}. \tag{2}$$

Here, $P = \begin{bmatrix} R & T \\ \mathbf{0}^T & 1 \end{bmatrix}$ represents the old position of the camera or the target object. $\hat{P}_i$ is the new relative camera position.

### 3.3 REPRESENTATION OF STREET SCENES

In a self-driving system (*e.g.* Geosim Chen et al. (2021b)), vehicles on the road play the most important role in driving decisions. There are some important application scenarios (*e.g.* expressway) only contain cars. In this paper, we mainly focus on the background scenes and moving vehicles. Other rigid and nonrigid objects can also use the similar techniques.

**Background scenes** The street view sequences in nuScenes and Waymo datasets usually span a long range (>200m), so it is necessary to constrain the whole scene into a bounded range before the position encoding. The Normalized Device Coordinates (NDC) (Mildenhall et al., 2020) fails in our scenes due to the complicated motions of the ego car and the challenging camera settings (Figure 1(b)).

We improve the scene parameterization function used in Barron et al. (2022) as:

$$f(x) = \begin{cases} x/r, & \text{if } \|x\| \leq r, \\ \left(2 - \frac{r}{\|x\|}\right)\frac{x}{\|x\|}, & \text{otherwise.} \end{cases} \tag{3}$$

Where $r$ is the radius parameter to decide the mapping boundary. The mapping is independent of the far bounds and avoids the heavy range compressing of the close objects, which gives more details in rendering the close objects (controlled by $r$). We use frustum from Mip-NeRF (Barron et al., 2021), sampling along rays evenly in a log scale to get more points close to the near plane.

**Moving Vehicles** In order to train NeRF with a limited number of views (*e.g.* $2 \sim 6$ image views), we compute the dense depth maps for the moving cars as an extra supervision. We follow GeoSim (Chen et al., 2021b) to reconstruct coarse mesh from multi-view images and the sparse LiDAR points. After that, a differentiable neural renderer (Liu et al., 2019) is used to render the corresponding depth map with the camera parameter (Section 3.2). The backgrounds are masked during the training by an instance segmentation network (Wang et al., 2020).

### 3.4 DEPTH SUPERVISION

As illustrated in Figure 3, to provide credible depth supervisions from defect LiDAR depths, we first propagate the sparse depths and then construct a confidence map to address the depth outliers. Our depth confidence is defined as a learnable combination of the **reprojection confidence** and the **geometry confidence**. During the training, the confidence maps are jointly optimized with the color rendering for each input view.

**LiDAR depth completion**   We use NLSPN (Park et al., 2020) to propagate the depth information from LiDAR points to surrounding pixels. While NLSPN performs well with 64-channel LiDAR data (*e.g.* KITTI Geiger et al. (2012; 2013) dataset), it doesn't generate good results with nuScenes' 32-channel LiDAR data, which are too sparse for the depth completion network. As a result, we accumulate neighbour LiDAR frames to get much denser depths for NLSPN. These accumulated LiDAR data, however, contain a great quantity of outliers due to the moving objects, ill poses and occlusions, which give wrong depth supervisions. To address this problem, we design a robust confidence measurement that can be jointly optimized while training our S-NeRF.

**Reprojection confidence**   To measure the accuracy of the depths and locate the outliers, we first use an warping operation $\psi$ to reproject pixels $\mathbf{X} = (x, y, d)$ from the source images $I_s$ to the target image $I_t$. Let $P_s$, $P_t$ be the source and the target camera parameters, $d \in \mathcal{D}_s$ as the depth from the source view. The warping operation can be represented as:

$$\mathbf{X}_t = \psi(\psi^{-1}(\mathbf{X}_s, P_s), \ P_t) \tag{4}$$

$\psi$ represents the warping function that maps 3D points to the camera plane and $\psi^{-1}$ refers to the inverse operation from 2D to 3D points. Since the warping process relies on depth maps $\mathcal{D}_s$, the depth outliers can be located by comparing the source image and the inverse warping one. We introduce the RGB, SSIM (Wang et al., 2004) and the pre-trained VGG feature (Simonyan & Zisserman, 2015) similarities to measure the projection confidence in the pixel, patch structure, and feature levels:

$$\mathcal{C}_{\text{rgb}} = 1 - |\mathcal{I}_s - \hat{\mathcal{I}}_s|, \quad \mathcal{C}_{\text{ssim}} = \text{SSIM}(\mathcal{I}_s, \hat{\mathcal{I}}_s)), \quad \mathcal{C}_{\text{vgg}} = 1 - ||\mathcal{F}_s - \hat{\mathcal{F}}_s||. \tag{5}$$

Where $\hat{\mathcal{I}}_s = \mathcal{I}_t(\mathbf{X_t})$ is the warped RGB image, and the $\hat{\mathcal{F}}_s = \mathcal{F}_t(\mathbf{X_t})$ refers to the feature reprojection. The receptive fields of these confidence maps gradually expand from the pixels and the local patches to the non-local regions to construct robust confidence measurements.

**Geometry confidence**   We further impose a geometry constrain to measure the geometry consistency of the depths and flows across different views. Given a pixel $\mathbf{X}_s = (x_s, y_s, d_s)$ on the depth image $\mathcal{D}_s$ we project it to a set of target views using equation 4. The coordinates of the projected pixel $\mathbf{X}_t = (x_t, y_t, d_t)$ are then used to measure the geometry consistency. For the projected depth $d_t$, we compute its consistency with the original target view's depth $\hat{d}_t = D_t(x_t, y_t)$:

$$\mathcal{C}_{depth} = \gamma(|d_t - \hat{d}_t|/d_s), \quad \gamma(x) = \begin{cases} 0, & \text{if } x \geq \tau, \\ 1 - x/\tau, & \text{otherwise.} \end{cases} \tag{6}$$

For the flow consistency, we use the optical flow method (Zhang et al., 2021) to compute the pixel's motions from the source image to the adjacent target views $f_{s \rightarrow t}$. The flow consistency is then formulated as:

$$\mathcal{C}_{flow} = \gamma(\frac{\|\Delta_{x,y} - f_{s \rightarrow t}(x_s, y_s)\|}{\|\Delta_{x,y}\|}), \quad \Delta_{x,y} = (x_t - x_s, y_t - y_s). \tag{7}$$

Where $\tau$ is a threshold in $\gamma$ to identify the outliers through the depth and flow consistencies.

**Learnable confidence combination**   To compute robust confidence map, we define the learnable weights $\omega$ for each individual confidence metric and jointly optimize them during the training. The final confidence map can be learned as $\hat{\mathcal{C}} = \sum_i \omega_i \mathcal{C}_i$, where $\sum_i \omega_i = 1$ The $i \in \{rgb, ssim, vgg, depth, flow\}$ represents the optional confidence metrics. The learnable weights $\omega$ adapt the model to automatically focus on correct confidence.

## 3.5   LOSS FUNCTIONS

Our loss function consists of a RGB loss that follows (Mildenhall et al., 2020; Barron et al., 2021; Deng et al., 2022), and a confidence-conditioned depth loss. To boost the depth learning, we also employ edge-aware smoothness constraints, as in (Li et al., 2021a; Godard et al., 2019), to penalize large variances in depth according to the image gradient $|\partial I|$:

$$\mathcal{L}_{\text{color}} = \sum_{\mathbf{r} \in \mathcal{R}} \|I(\mathbf{r}) - \hat{I}(\mathbf{r})\|_2^2 \tag{8}$$

$$\mathcal{L}_{depth} = \sum \hat{\mathcal{C}} \cdot |\mathcal{D} - \hat{\mathcal{D}}| \tag{9}$$

$$\mathcal{L}_{smooth} = \sum |\partial_x \hat{\mathcal{D}}| \ \exp^{-|\partial_x I|} + |\partial_y \hat{\mathcal{D}}| \ \exp^{-|\partial_y I|} \tag{10}$$

$$\mathcal{L}_{\text{total}} = \mathcal{L}_{\text{color}} + \lambda_1 \mathcal{L}_{depth} + \lambda_2 \mathcal{L}_{smooth} \tag{11}$$

| Methods | Static Vehicles | | | Moving Vehicles | | |
|---|---|---|---|---|---|---|
| | PSNR↑ | SSIM↑ | LPIPS↓ | PSNR↑ | SSIM↑ | LPIPS↓ |
| NeRF (Mildenhall et al., 2020) | 11.78 | 0.539 | 0.444 | – | – | – |
| GeoSim (Chen et al., 2021b) | 11.58 | 0.602 | 0.367 | 12.24 | 0.623 | 0.322 |
| Ours | **18.81** | **0.785** | **0.194** | **18.00** | **0.736** | **0.226** |

Table 1: Novel view synthesis results on foreground cars. We compare our method with the NeRF and GeoSim baselines. Since COLMAP fails on foreground vehicles, we apply our camera parameters to the NeRF baseline when training the static vehicles. We report the quantitative results on PSNR, SSIM (Wang et al., 2004) (higher is better) and LPIPS (Zhang et al., 2018) (lower is better).

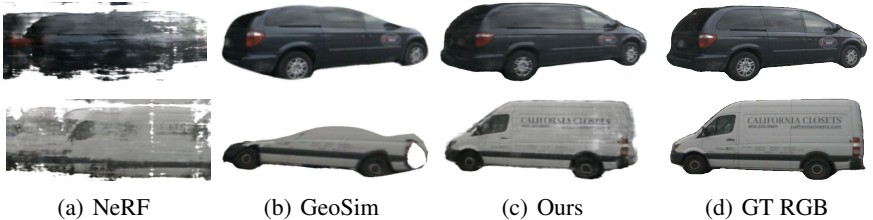

| (a) NeRF | (b) GeoSim | (c) Ours | (d) GT RGB |

Figure 5: Novel-view synthesis results for static foreground vehicles. Results are reconstructed from 4∼7 views. Our method outperforms others (Chen et al., 2021b; Mildenhall et al., 2020) with more texture details and accurate shapes.

Where $\mathcal{R}$ is a set of rays in the training set. For the reconstruction of the foreground vehicles, $\mathcal{D}$ refers to the depth. And in background scenes, $\mathcal{D}$ represents the disparity (inverse depth) to make the model focus on learning important close objects. $\lambda_1$ and $\lambda_2$ are two user-defined balance weights.

## 4 EXPERIMENTS

We perform our experiments on two open source self-driving datasets: nuScenes (Caesar et al., 2019) and Waymo (Sun et al., 2020). We compare our S-NeRF with the state-of-the-art methods (Barron et al., 2021; 2022; Rematas et al., 2022; Chen et al., 2021b). For the foreground vehicles, we extract car crops from nuScenes and Waymo video sequences. For the large-scale background scenes, we use scenes with 90∼180 images. In each scene, the ego vehicle moves around 10∼40 meters, and the whole scenes span more than 200m. We do not test much longer scenes limited by the single NeRF's representation ability. Our model can merge different sequences like Block-NeRF (Tancik et al., 2022) and achieve a larger city-level representation.

In all the experiments, the depth and smooth loss weight $\lambda_1$ and $\lambda_2$ are set to 1 and 0.15 respectively for foreground vehicles. And for background street scenes, we set $\tau = 20\%$ for confidence measurement and the radius $r = 3$ in all scenes. $\lambda_1 = 0.2$ and $\lambda_2 = 0.01$ are used as the loss balance weights. More training details are available in the supplementary materials.

### 4.1 NOVEL VIEW RENDERING FOR FOREGROUND VEHICLES

In this section, we present our evaluation results for foreground vehicles. We compare our method with the latest non-NeRF car reconstruction method (Chen et al., 2021b). Note that existing NeRFs (Mildenhall et al., 2020; Deng et al., 2022) cannot be used to reconstruct the moving vehicles. To compare with the NeRF baseline (Mildenhall et al., 2020), we also test our method on the static cars. Since COLMAP (Schönberger & Frahm, 2016) fails in reconstruct the camera parameters here, the same camera poses used by our S-NeRF are applied to the NeRF baseline to implement comparisons.

**Static vehicles** Figure 5 and Table 1 show quantitative and visualized comparisons between our method and others in novel view synthesis. Optimizing NeRF baseline on a few (4∼7) image views leads to severe blurs and artifacts. GeoSim produces texture holes when warping textures for novel view rendering. The shapes of the cars are also broken due to the inaccurate meshes. In contrast, our S-NeRF shows more fine texture details and accurate object shapes. It can improve the PSNR and SSIM by 45∼65% compared with the NeRF and GeoSim baselines.

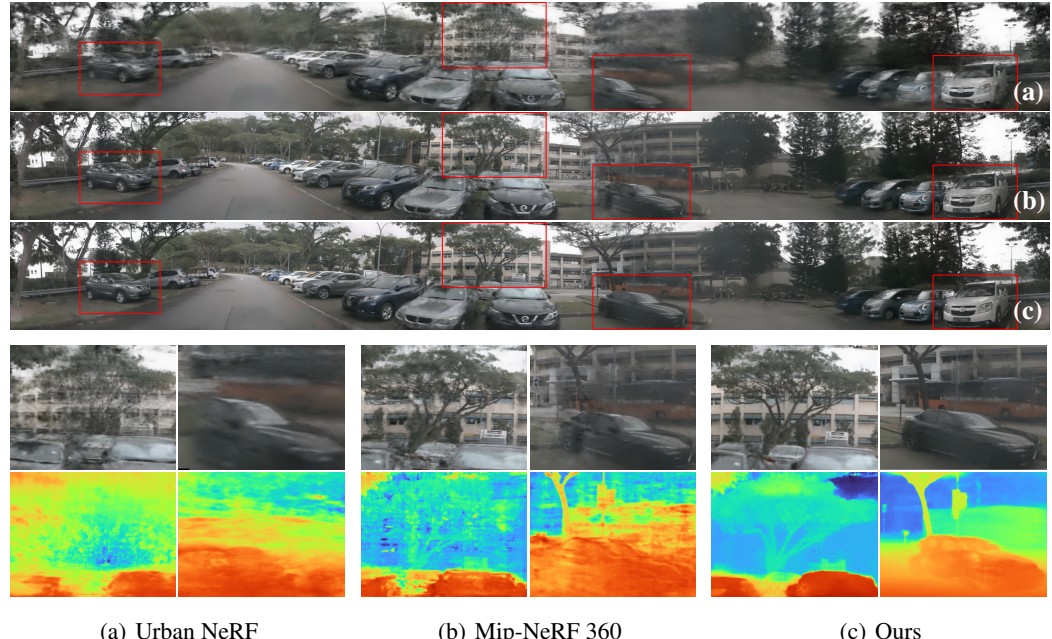

(a) Urban NeRF        (b) Mip-NeRF 360        (c) Ours

Figure 6: We render the 360 degree panoramas (in a resolution of $8000 \times 800$) for comparisons (*zoom in to see details*). Significant improvements are highlighted by red rectangles and cropped patches are shown to highlight details. *See the appendix for more results.*

| Large-scale Scenes Synthesis | | | |
|---|---|---|---|
| Methods | PSNR↑ | SSIM↑ | LPIPS↓ |
| Mip-NeRF (Barron et al., 2021) | 18.22 | 0.655 | 0.421 |
| Mip-NeRF360 (Barron et al., 2021) | 24.37 | 0.795 | 0.240 |
| Urban-NeRF (Rematas et al., 2022) | 21.49 | 0.661 | 0.491 |
| Ours | **26.21** | **0.831** | **0.228** |

Table 2: Our method quantitatively outperforms state-of-the-art methods. Methods are tested on four nuScenes Sequences. Average PSNR, SSIM and LPIPS are reported.

**Moving vehicles** As compared in Figure 2(c)∼2(d) and Table 1, novel view synthesis by GeoSim (Chen et al., 2021b) is sensitive to mesh errors that will make part of the texture missing or distorted during novel view warping and rendering. In contrast, our S-NeRF provides larger ranges for novel view rendering than geosim (see supplementary) and generates better synthesis results. S-NeRF can also simulate the lighting changing for different viewing directions, which is impossible for GeoSim. Furthermore, S-NeRF does not heavily rely on accurate mesh priors to render photorealistic views. The confidence-based design enables S-NeRF to eliminate the geometry inconsistency caused by depth outliers. S-NeRF surpasses the latest mesh based method (Chen et al., 2021b) by 45% in PSNR and 18% in SSIM.

## 4.2 NOVEL VIEW RENDERING FOR BACKGROUND SCENES

Here we demonstrate the performance of our S-NeRF by comparing with the state-of-the-art methods Mip-NeRF (Barron et al., 2021), Urban-NeRF (Rematas et al., 2022) and Mip-NeRF 360 (Barron et al., 2022). We use the offical code of Mip-Nerf for evaluation and expand the hidden units of the MLP to 1024 (the same as ours). Since there is no official code published, we tried our best to reproduce (Rematas et al., 2022; Barron et al., 2022) based on our common settings. We test four nuScenes sequences and report the evaluation results in Table 2. Our method outperforms Mip-NeRF, Mip-NeRF 360 and Urban NeRF in all three evaluation metrics. We see significant improvements of 40% in PSNR and 26% in SSIM and a 45% reduction in LPIPS compared with the Mip-NeRF baseline. It also outperforms the current best Mip-NeRF 360 by 7.5% in PSNR, 4.5% in SSIM and 5% in LPIPS. We also show 360-degree panoramic rendering in Figure 6. S-NeRF significantly ameliorates artifacts, suppresses "floaters" and presents more fine details compared with Urban-NeRF and Mip-NeRF 360. *Results of Waymo scenes are shown in the appendix.*

| | Background Scenes (background) | | | Moving Vheciles | | |
|---|---|---|---|---|---|---|
| | PSNR↑ | SSIM↑ | LPIPS↓ | PSNR↑ | SSIM↑ | LPIPS↓ |
| Mip-NeRF or GeoSim | 15.55 | 0.533 | 0.47 | 13.00 | 0.651 | 0.280 |
| Our RGB | 24.41 | 0.790 | **0.230** | 15.26 | 0.718 | 0.282 |
| w/o depth confidence | 24.30 | 0.775 | 0.279 | 18.09 | 0.748 | 0.206 |
| w/o smooth loss | **25.06** | 0.804 | 0.231 | 19.62 | 0.796 | 0.159 |
| Full Settings | 25.01 | **0.805** | 0.232 | **19.64** | **0.803** | **0.136** |

Table 3: Ablations on different settings. For background scenes, we use two nuScenes sequences for evaluations. For moving vehicles, four cars are trained under different settings.

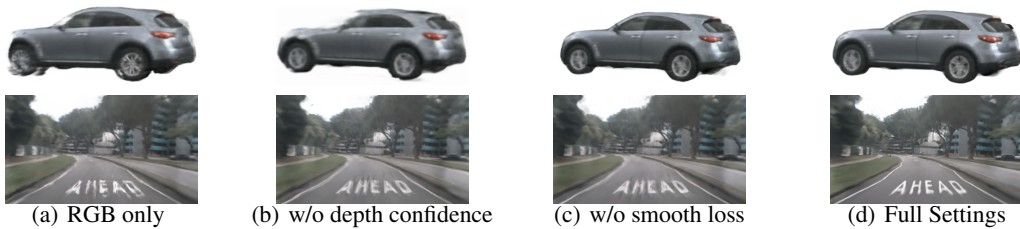

(a) RGB only   (b) w/o depth confidence   (c) w/o smooth loss   (d) Full Settings

Figure 7: Ablation study on different training settings of (a) RGB only, (b) noisy depth supervision, (c) using depth confidence, and (d) full supervision settings.

## 4.3 BACKGROUND AND FOREGROUND FUSION

There are two different routes to realize controllable foreground and background fusion: depth-guided placement and inpainting (*e.g.* GeoSim Chen et al. (2021b)) and joint NeRF rendering (*e.g.* GIRAFFE Niemeyer & Geiger (2021)). Both of them heavily rely on accurate depth maps and 3D geometry information for object placement, occlusion handling, *etc.*. Our method can predict far better depth maps and 3D geometry (*e.g.* meshes of cars) than existing NeRFs (Zhang et al., 2020; Barron et al., 2022). We provide a video (in the supplementary materials) to show the controlled placement of vehicles to the background using the depth-guided placement.

## 4.4 ABLATION STUDY

To further demonstrate the effectiveness of our method, we conduct a series of ablation studies with or without certain component. Table 3 shows the quantitative results of our model ablations. We evaluate our model on four foreground vehicles and two background scenes under different settings. Visualized results are also provided in Figure 7. *More ablation studies are available in the appendix.*

For background scene rendering, our RGB baseline outperforms the Mip-NeRF by 56% in mean-squared error and 42% in structure similarity. Using inaccurate depth supervision without confidence leads to a drop of the accuracy due to the depth outliers. Confidence contributes to about 3% improvement in PSNR and SSIM. PSNR of our model slightly drops after adding the edge-aware smooth loss to it. However, it effectively suppresses "floaters" and outliers to improve our depth quality. For moving vehicles, the depth supervision and confidence measurement improves the RGB baseline by 18% and 8% in PSNR individually. The smoothness loss mainly improves the structure similarity and reduces the LPIPS errors.

## 5 CONCLUSION, LIMITATIONS AND FUTURE WORK

We contribute a novel NeRF design for novel view rendering of both the large-scale scenes and foreground moving vehicles using the steet view datasets collected by self-driving cars. In the experiments, we demonstrate that our S-NeRF far outperforms the state-of-the-art NeRFs with higher-quality RGB and impressive depth renderings. Though S-NeRF significantly outperforms Mip-NeRF and other prior work, it still produces some artifacts in the depth rendering. For example, it produces depth errors in some reflective windows. In the future, we will use the block merging as proposed in Block-NeRF (Tancik et al., 2022) to learn a larger city-level neural representation.

**Acknowledgments** This work was supported in part by National Natural Science Foundation of China (Grant No. 62106050), Lingang Laboratory (Grant No. LG-QS-202202-07), Natural Science Foundation of Shanghai (Grant No. 22ZR1407500).

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

# APPENDIX

## A  MORE ILLUSTRATIONS

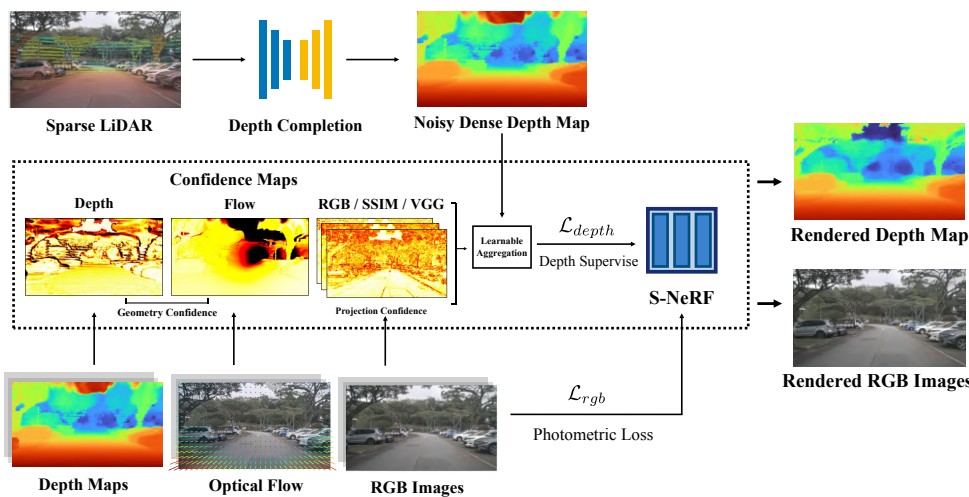

Figure 8: Overview of our S-NeRF framework.

**Method Overview**   The overview of the method is shown in Figure 8. We first propagate the sparse LiDAR points into a dense depth map and compute the geometry and the projection confidence maps. We use learnable combination to achieve the final confidence maps to reduce the influence of depth outliers.

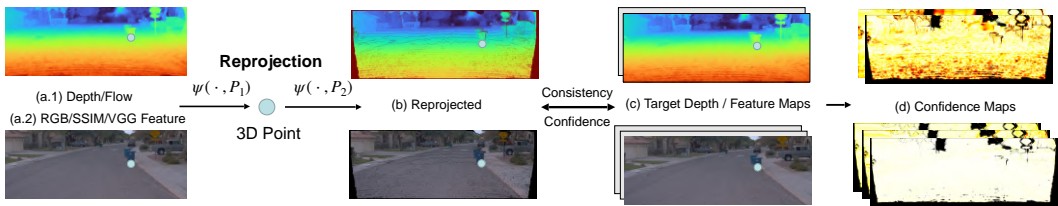

Figure 9: Illustration of the confidence computation.

**Computation of Confidence**   Figure 9 shows the process of computing confidence maps. We compute the depth confidence by reprojection the depth maps to other views. Given the LiDAR positions of the consecutive frames $[P_{t-1}, P_t, P_{t+1}]$ at different time $[t-1, t, t+1]$, we are able to project the 3D LiDAR points at time t-1 and t+1 to t by $\Delta P$ between $P_t$ and $P_{t-1}/P_{t+1}$. This is similar to the mapping function $\psi$ in Eq. (4) of the paper. Most outliers of the moving objects and other regions can be removed by the consistency check using the optical flow and the depth projection (Eq. (6-7) of the paper). The rest outliers can also be handled by the proposed confidence-guided learning.

**Illustration of Confidence Maps**   In Figure 10, we visualize different confidence components. Depth and optical flow confidence maps focus on geometry consistency between adjacent frames, while RGB, SSIM and VGG confidence maps compute the photometric and feature consistency.

---

Images in the appendix are slightly compressed. Lossless images are available at https://ziyang-xie.github.io/s-nerf.

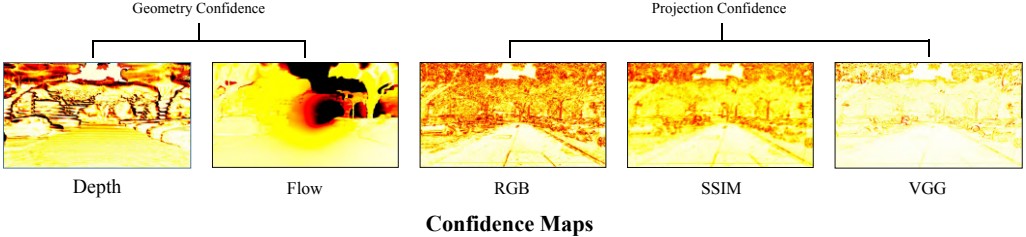

**Confidence Maps**

Figure 10: Visualization of each confidence component. Brighter regions mean higher confidence. Geometry confidences (flow and depth) represents the geometry consistency as computed in Fig. 2(a.1). The projection confidence measure the photometric and feature consistency as computed in Fig. 2(a.2).

**Other Dynamic Objects.** In Fig. 11, we reconstruct a moving truck using only 4 image views. The novel-view rendering quality (Fig. 11) is good enough for our driving simulation. For the dynamic person, since only a limited number of image views can be captured for a single person in the nuScenes and Waymo datasets, and the person is also walking with varying poses. It's difficult to reconstruct high-quality 3D person using only a few (2-5) views. We instead use existing monocular video data to reconstruct 3D person and rendering novel views with novel poses. Fig. 12 shows examples using Anim-NeRF Chen et al. (2021a) to reconstruct persons in People-Snapshot dataset Alldieck et al. (2018). Then, we can put the 3D persons (with novel views and novel poses) to the S-NeRF scenes for the realistic driving simulation (Fig. 13). Similarly, other objects (e.g. bicycles) can be first reconstructed and then combined into our S-NeRF scenes for driving simulation.

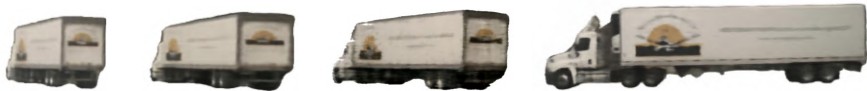

Figure 11: Novel view rendering of a reconstructed moving truck.

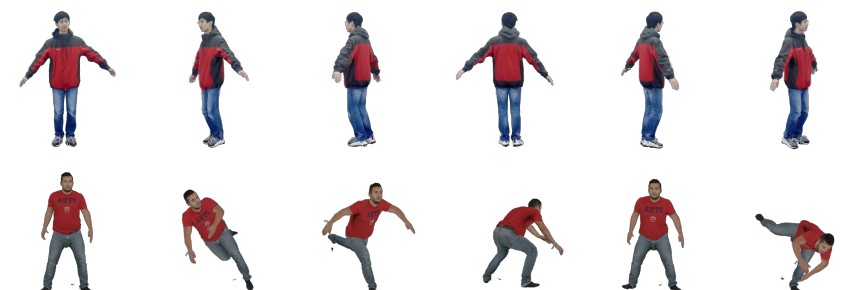

Figure 12: Novel-view and novel-pose rendering for the reconstructed dynamic persons.

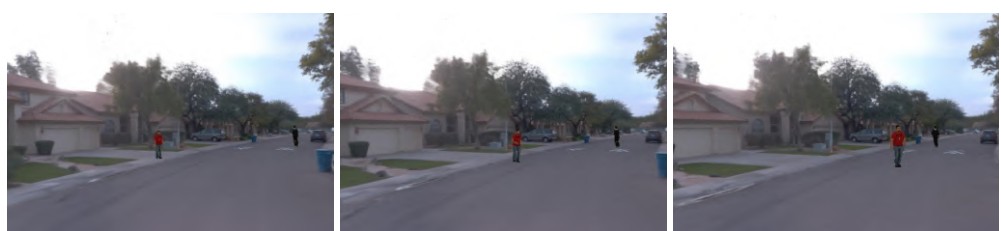

Figure 13: After novel-view and novel-pose rendering for the reconstructed persons, we also combine the dynamic persons into our rendered S-NeRF scenes.

# B  IMPLEMENTATION DETAILS

## B.1  DEPTH COMPLETION

We use NLSPN[Park et al. (2020)] network for depth completion, which propagates the depth information from sparse LiDAR points to surrounding pixels. While NLSPN performs well with 64-channel LiDAR data, such as KITTIGeiger et al. (2012; 2013) and waymoSun et al. (2020); Ettinger et al. (2021) datasets, it doesn't generate good results with nuScenesCaesar et al. (2019) 32-channel LiDAR data, which are too sparse to complete the depth. We accumulate 5∼10 neighbour LiDAR frames to get a much denser LiDAR data for the NLSPN network. These accumulated LiDAR points, however, contains many outliers due to the moving objects, ill poses, occlusions and reprojection errors. These outliers will give wrong depth information. To remove these outliers, we first compute the optical flow Zhang et al. (2021) of one RGB image using its neighbour frame. After that, we reproject each LiDAR points to neighbour image plane to get the LiDAR flow, and then compare two kinds of flows to locate the LiDAR outliers using a threshold of 20% (following Eq. (6∼8) of the paper).

Applying above procedure, we can remove many outliers. However, some of these outliers still exists due to the errors of the optical flow and ill poses. These outliers still exist after depth completion. The depth completion algorithm also introduces new outliers to the final dense depth map, which are great challenges for depth supervision. We therefore learn an confidence metric for more robust depth supervision.

## B.2  REMOVE MOVING OBJECTS IN STREET VIEWS

Currently, we focus on static scenes. When training the background scenes, we masked the moving objects, while static objects (*e.g.*static vehicles) are kept and trained along with the background. The moving vehicles are trained independently. Other moving objects (*e.g.*person) can be removed by instance segmentation and optical flow.

## B.3  FOREGROUND VEHICLES

For foreground vehicles, we use four layers with 256 hidden units in the MLP. The depth and smooth loss weights $\lambda_1$ and $\lambda_2$ are set to 1 and 0.15 respectively. We sample 64 times along each ray during the RGB and depth rendering. We train our S-NeRF for 30k iterations using Adam optimizer with $5^{-4}$ as the learning rate and 1024 as the batch size. The training takes about 2 hours for each vehicle on a single RTX3090 gpu. For each vehicle, there are around 2∼8 views used for training and 1∼3 views for testing.

## B.4  STREET VIEWS (BACKGROUND)

For background street scenes, we set $\tau = 20\%$ for the confidence measurement and the radius $r = 3$ for the scene parameterization function. Our coarse-fine network share the same parameters like mip-NeRF. The density MLP has eight layers with 1024 hidden units and the color MLP consists of three layers with 128 hidden units. We keep this setting for all evaluation methods for fair comparisons. We train our S-NeRF for 100K iterations using Adam optimizer with a batch size of 2048. The learning rate is reduced log-linearly from $5\times 10^{-4}$ to $5\times 10^{-6}$ with a warm-up phase of 2500 iterations. $\lambda_1 = 0.2$ and $\lambda_2 = 0.01$ are used as the loss balance weights. We sample 128 times along each ray in a log scale. Our S-NeRF is trained on two RTX3090 GPUs which takes about 17 hours for a scene with about 250 images (with a resolution of 1280×1920).

# C  EXPERIMENTS

We present more visualized results in Figure 14 and Figure 16. We also provide a video demo for performance illustration.

| Large-scale Scenes Synthesis on Waymo dataset | | | |
|---|---|---|---|
| Methods | PSNR↑ | SSIM↑ | LPIPS↓ |
| Mip-NeRF Barron et al. (2021) | 16.89 | 0.412 | 0.755 |
| Mip-NeRF360 Barron et al. (2021) | 22.10 | 0.724 | 0.445 |
| Urban-NeRF Rematas et al. (2022) | 17.80 | 0.494 | 0.701 |
| Ours | **23.60** | **0.743** | **0.422** |

Table 4: Our method quantitatively outperforms state-of-the-art methods Barron et al. (2021; 2022); Rematas et al. (2022). Methods are trained on two waymo scenes. Average PSNR, SSIM and LPIPS are reported.

| Foreground Vehicles | | | |
|---|---|---|---|
| Methods | PSNR↑ | SSIM↑ | LPIPS↓ |
| NeRF Mildenhall et al. (2020) | 14.22 | 0.739 | 0.32 |
| GeoSim Chen et al. (2021b) | 14.27 | 0.742 | 0.186 |
| Ours | **23.16** | **0.870** | **0.156** |

Table 5: Novel view synthesis results on foreground cars from Waymo dataset. We compare our method with the NeRF and GeoSim baselines. Since COLMAP fails on foreground vehicles, we apply our camera parameters to the NeRF baseline when training the static vehicles. We report the quantitative results on PSNR, SSIM Wang et al. (2004) (higher is better) and LPIPS Zhang et al. (2018) (lower is better).

## C.1 Parameters and Efficiency

In the experiennts, we use the same settings for the MLP encoding in our S-NeRF and other state-of-the-art methods Barron et al. (2021; 2022); Rematas et al. (2022). There are 8.76M learnable parameters in our S-NeRF that is similar to other state-of-the-art methods Barron et al. (2021; 2022); Rematas et al. (2022) (8.7∼9.9M).

All the methods are trained for 100k iterations. Our method takes about 17 hours in training for one street scene. This is the same as Mip-NeRF and Urban-NeRF because we use the same settings during the experiments. Mip-NeRF 360 is faster than ours because it doesn't require the coarse rendering for supervision. This strategy can also be used in our S-NeRF to accelerate the convergence and further improve the quality of the novel view rendering.

## C.2 Waymo Results

We also test our S-NeRF on two Waymo street-view sequences. The results are reported in Table 4 and visually compared with the state-of-the-arts Rematas et al. (2022); Barron et al. (2022) in Figure 15 and 16. Waymo dataset use a 64-channel LiDAR and five cameras for capture driving scenes. Our S-NeRF outperforms Mip-NeRF by 40% in PSNR, 80% in SSIM and 44% in LPIPS. It's also far better than the Urban-NeRF baseline (32∼50% ↑ in PSNR, SSIM and LPIPS) which also uses the sparse LiDAR depth as supervision. Compared with the current best Mip-NeRF 360, our method also achieves a 6.8% improvements in PSNR and a 5.2% reduction of the LPIPS error. More importantly, our S-NeRF predicts much more accurate depths for the large-scale street views.

We also use Waymo dataset to compare our method with the NeRF baseline on the vehicle synthesis. The quantitive evaluations are shown in Table 5 and visually compared in Figure 14. Our method outperforms the NeRF baseline by 63% in PSNR, 18% in SSIM and 51% in LPIPS. Compared with the recently proposed mesh-based car reconstruction method Chen et al. (2021b), our S-NeRF improves the PSNR by 62% and reduces the LPIPS error by 16%.

## C.3 Comparisons with Urban-NeRF

Urban-NeRF also use the LiDAR depth to boost the NeRF training. However, it requires accurate LiDAR depth. Most of the street view datasets (*e.g.* Waymo and nuScenes) contains plenty of depth

| $\lambda_1$ | $\lambda_2$ | PSNR↑ | SSIM↑ | LPIPS↓ |
|---|---|---|---|---|
| 0.1 | 0.01 | 24.00 | 0.770 | 0.385 |
| 0.4 | 0.01 | 23.85 | 0.767 | 0.397 |
| 0.2 | 0.005 | 23.95 | 0.769 | **0.382** |
| 0.2 | 0.02 | 23.98 | **0.771** | 0.390 |
| 0.2 | 0.05 | 23.78 | 0.767 | 0.406 |
| 0.2 | 0.01 | **24.05** | **0.771** | 0.384 |

Table 6: Ablations on loss balance weights $\lambda_1$, $\lambda_2$ for background street views.

| $\lambda_1$ | $\lambda_2$ | PSNR↑ | SSIM↑ | LPIPS↓ |
|---|---|---|---|---|
| 0.5 | 0.15 | 19.56 | 0.794 | 0.149 |
| 2 | 0.15 | 19.67 | 0.802 | 0.149 |
| 1 | 0.075 | **19.85** | **0.803** | 0.147 |
| 1 | 0.3 | 19.47 | 0.789 | 0.147 |
| 1 | 0.15 | 19.64 | **0.803** | **0.136** |

Table 7: Ablations on loss balance weights $\lambda_1$, $\lambda_2$ for foreground vehicles.

outliers due to the influence of ill poses, reprojection errors, occlusion and moving objects. We find that these depth outliers can heavily influence the rendering quality of Urban-NeRF. Our S-NeRF introduces confidence metrics to make it more robust to the depth outliers. Moreover, our S-NeRF improves the scene parameterization function and the camera poses to learn better neural representation for rendering large-scale street views. As shown in Figure 16 and 15, compared with Urban-NeRF, our S-NeRF gives more details in street-view objects (such as the car, the tree and road lines) and are more robust for the large-scale street views captured by the self-driving cars.

### C.4 MORE ABLATION STUDY

In this section, we use two nuScenes street-view scenes and three foreground vehicles to test the effects of different settings when training our S-NeRF. These include different loss balance weights, different confidence components, different depth qualities and different scene parameterizations. In these ablation experiments, S-NeRF is trained for 30k iterations.

**Loss balance weights**

As shown in Table 6 and 7, we compare the performance of our S-NeRF using different loss balance weights. Our S-NeRF is not very sensitive to the changes of the loss balance weights. Using larger or smaller $\lambda_1$ or $\lambda_2$ just slightly reduces the PSNR and SSIM by 0.2~1% on the background novel view rendering. $[0.1, 0.4]$ (for $\lambda_1$) and $(0.005, 0.02)$ (for $\lambda_2$) are the reasonable range for our loss balance weights for training street background views. For the foreground vehicles, $[0.5, 2]$ (for $\lambda_1$) and $(0.075, 0.015)$ (for $\lambda_2$) are the reasonable range.

**Confidence components**

We also test the performances of our S-NeRF when using different confidence components (geometry and reprojection measurements) for learning our confidence metric. As shown in Table 8, when we remove the reprojection confidence module, the PSNR slightly dropped by 0.4%. And when we train S-NeRF without geometry confidence, the PSNR and SSIM is about 0.7% lower. We also test the effects of the threshold $\tau$ used in the geometry confidence (Eq. (7) of the paper). We find that the geometry confidence is not sensitive to the threshold $\tau$. $[10\%, 40\%]$ is a reasonable range for the threshold $\tau$.

| Reporjection | Geometry | PSNR↑ | SSIM↑ | LPIPS↓ |
|:---:|:---:|:---:|:---:|:---:|
| Yes | No | 23.88 | 0.767 | 0.385 |
| No | Yes ($\tau = 20\%$) | 23.95 | 0.770 | 0.385 |
| Yes | Yes ($\tau = 10\%$) | 23.93 | **0.771** | 0.385 |
| Yes | Yes ($\tau = 40\%$) | 23.97 | **0.771** | **0.384** |
| Yes | Yes ($\tau = 20\%$) | **24.05** | **0.771** | **0.384** |

Table 8: Ablations on confidence settings for novel street-view rendering (background)..

| RGB confidence | SSIM confidence | PSNR↑ | SSIM↑ | LPIPS↓ |
|:---:|:---:|:---:|:---:|:---:|
| Yes | No | 19.79 | 0.803 | 0.152 |
| No | Yes | **19.97** | **0.807** | 0.145 |
| Yes | Yes | 19.64 | 0.803 | **0.136** |

Table 9: Ablations on confidence components for foreground vehicles.

For the foreground vehicles, we only use RGB and SSIM confidences. This is because the depth map used in training vehicles are relatively better than the backgrounds. Thus, we do not need strong geometry confidences. We report these ablation results in Table 9 by using only RGB confidence or SSIM confidence. We find that using only RGB or SSIM confidence could achieve a little better PSNR (1∼2%↑), but a relative worse LPLPS (6.6 ∼ 11%↓). Taken all these three evaluation metrics into consideration, using both RGB and SSIM confidence gives a better performance in training our S-NeRF. We also report depth error rate in Table 10. For the accurate depths, it predicts high confidence to encourage the NeRF geometry to be consistent with the LiDAR signal.

**Depth quality**

As shown in Table 12, we study the effects of different depth map qualities. We train our S-NeRF using depth map in different qualities. To simulate the depth errors in different qualities, we add random Gaussian Noise to the original depth map inputs. The strength of the noise (the quality of the depths) is measured by PSNR and error rates compared with the original depth inputs. Error rate means how many outliers are introduced by the noise. We use $threshold = 1$ to compute the outlier rates compared with the original depth inputs. Our method achieve similar rendering qualities when training with light noises, which means the light noises doesn't influence the performance of our S-NeRF. When training with strong noises, the PSNR and SSIM just slightly dropped by 0.05∼0.5%. This shows that our method is robust to depth noises because our confidence strategy can locate and measure the depth outliers accurately and avoid the negative influence of the depth noises in training our S-NeRF.

Besides, we also test the performance of our S-NeRF when using only sparse depth for supervision (Table 11). It performs better than our RGB-only S-NeRF (improving the PSNR by 5%). But, it is 2% worse in PSNR and 8% worse in LPIPS than our default settings where dense depth map and the proposed confidence metric are used. This means that dense depth maps can provide more useful geometry information for training our S-NeRF even though they may contain more depth outliers.

In addition, we also test another two worse depth completion methods by replacing the NLSPN with Ku et al. (2018); Van Gansbeke et al. (2019)(Table 13). Even using the worse traditional depth completion method Ku et al. (2018)(without learning a deep neural network), our S-NeRF still achieves a similar rendering quality (24.41→24.40). This experiment shows that our method doesn't rely on NLSPN depth quality. Benefiting from our confidence-guided depth supervision, many other depth completion methods can also be used in our S-NeRF.

**Scene and ray parameterization**

| Components of confidence | | | | | Metrics | |
| Rerojection confidence | | | Geometry confidence | | | |
| rgb | ssim | vgg | depth | flow | PSNR | Depth error rate |
|---|---|---|---|---|---|---|
| | | | | | 23.15 | 30.70% |
| ✓ | | | | | 23.83 | 26.85% |
| | ✓ | | | | 23.97 | 26.73% |
| | | ✓ | | | 23.94 | 26.25% |
| | | | ✓ | | 23.95 | 25.50% |
| | | | | ✓ | 24.03 | 24.92% |
| | | | ✓ | ✓ | 23.95 | 24.59% |
| ✓ | ✓ | ✓ | | | 23.88 | 26.75% |
| ✓ | ✓ | ✓ | ✓ | ✓ | **24.05** | **24.43%** |

Table 10: Ablation study on confidence components.

| Depth Settings | PSNR↑ | SSIM↑ | LPIPS↓ |
|---|---|---|---|
| No Depth | 22.45 | 0.742 | 0.433 |
| Sparse | 23.65 | 0.756 | 0.415 |
| Dense w/o confidence | 23.15 | 0.757 | 0.424 |
| Dense w/ confidence | **24.05** | **0.771** | **0.384** |

Table 11: Ablations on depth supervision for novel street-view rendering (background).

We studied the effects of using different radius in the scene parameterization function (Eq. (2) of the paper). Radius $r$ is used to constrain the whole large-scale scene into a bounded range. Close and far points are parameterized by different distance mapping function to make our S-NeRF able to keep more details for the close objects. This is controlled by the radius parameter. In Table 14, we report the performance of our S-NeRF when using different radius for scene parameterization. We find that choosing $r$ in 3∼10m could produce a better results than the original setting in Mip-Nerf 360 Barron et al. (2022). This is because the street views in nuScenes and Waymo datasets usually span a long range (>200m). While, Mip-Nerf 360 Barron et al. (2022) do not have $r$ to adjust the scene parameterization as ours.

**Influence of quality of 3D detection results** As shown in Table 15, the quality of 3D object detector does not significantly influence the rendering quality. This is because we use pose refinement to improve the inaccurate initial results of the virtual camera pose estimation. The pose refinement is guided by depth supervision and visual multiview constraints in training our S-NeRF. We test the rendering results by using 3D bounding boxes of different qualities (mIoU: 0.79, 0.67 and 0.55) and compare them with the ground truth bounding boxes (achieved from the dataset labels). We find that the PSNR doesn't drop significantly compared with the ground truth bounding boxes. When the mIoU of the bounding boxes is 0.79 (using the default detector), the PSNR just slightly drops by 0.23, the SSIM drops by 0.04, and the LPIPS doesn't change. When we use poor detection results, our S-NeRF still produces good results with a slight drop in the rendering quality. (E.g. 0.49 ↓when the mIoU is 0.67 and 0.97 ↓for a poor mIoU of 0.55). We observed that the detected bounding boxes ( > 0.5 in IoU) are good enough for training our S-NeRF. This can be easily realized by many existing 3D object detectors.

**Different pose refinements** We have tried different pose refinements like NeRF–, BARF and SC-NeRF. As shown in Table 17, We find that NeRF– helps achieve the best quality when training on the self-driving dataset. Possibly because NeRF– is more straightforward. It directly learns shifts to the translation and rotation. The shifts are relatively easier to learn under depth supervision. It's

| Depth noise (PSNR) | Depth error rate (%) | PSNR↑ | SSIM↑ | LPIPS↓ |
|---|---|---|---|---|
| >50 | 9.5 | **24.07** | **0.772** | **0.384** |
| >50 | 15.2 | 24.04 | 0.771 | **0.384** |
| 50 | 26.5 | 24.01 | 0.771 | **0.384** |
| 35 | 85 | 23.97 | 0.770 | 0.386 |
| 25 | 95 | 23.95 | 0.767 | 0.391 |
| raw depth | 0 | 24.05 | 0.771 | **0.384** |

Table 12: We study the effects when training our S-NeRF with depth map in different qualities. We add random Gaussian Noise to the original input depth maps. The strength of the noise is measured as PSNR and error rates. Error rates represent how many outliers are introduced by the noise. Our S-NeRF is robust to depth noises.

| Method | Mean absolute error/[mm] (KITTI) | Rank on KITTI leaderboard | PSNR | SSIM | LPIPS |
|---|---|---|---|---|---|
| Traditional method Ku et al. (2018) | 302.60 | 116 | 24.40 | 0.782 | 0.344 |
| Van Gansbeke et al. (2019) | 215.02 | 61 | 24.50 | 0.785 | 0.344 |
| Our default NLSPN | 199.59 | 40 | 24.41 | 0.783 | 0.345 |

Table 13: Effects of different depth completion methods

also possible that such shifts are easier to estimate when there is a small overlapped field of view between different cameras (as illustrated in Figure 1 of the paper.)

**Training and Inference time**   Referring to Table 16, we show more details of our method against the existing large-scale NeRFs with the training time and inference time. Our method (default settings) uses the same training setting and network parameters as those of MipNeRF. They take the same time in training and inference. When we implement the MipNeRF-360 strategy (the distillation mode, all other settings are kept the same), the light proposal MLP reduces both the training time and the inference time. It runs 30% faster in training and 40% faster in inference than the default settings (with a drop of 0.33 in PSNR).

# D   LIMITATIONS AND SOCIAL IMPACTS

**Failure cases.**   Considering that the ego vehicle sometimes goes very fast, fewer images are captured by the side (left or right) cameras. Some objects/contents only appeared in one or two left/right images. This makes it hard to render high-quality left and right novel views. As shown in Table 18, since there are fewer views, the rendered side (left/right) views report worse PSNR, SSIM and LPIPS ( $5 \sim 12\% \downarrow$ ) compared with the front and back views. In the future, we will try some techniques (e.g. using a better method to densify depth or provide semantic supervision) to further improve the quality of rendering left and right views.

**Potential risks.**   Since our method can be used to render realistic street views and vehicles, it's possible that S-NeRF can be used to synthesize fake photos or videos. We therefore hope our S-NeRF could be used with cautiousness. Work that bases itself on our method should also carefully consider the consequences of this potential negative social influence.

| $r$ | PSNR↑ | SSIM↑ | LPIPS↓ |
|---|---|---|---|
| 1m | 23.95 | 0.769 | **0.384** |
| 5m | **24.07** | 0.771 | 0.385 |
| 10m | 24.05 | **0.771** | **0.384** |
| 3m | 24.05 | **0.771** | **0.384** |

Table 14: Ablations on radius $r$ in our scene parameterization function (EQ. (14) of the paper) for novel street-view rendering (background).

| Bounding box types | mIoU | PSNR | SSIM | LPIPS |
|---|---|---|---|---|
| GT bouding box | 1.0 | **23.50** | **0.862** | **0.111** |
| CenterPoints Yin et al. (2021) | 0.787 | 23.27 | 0.858 | **0.111** |
| poor bounding box | 0.672 | 23.01 | 0.845 | 0.126 |
| poor bounding box | 0.554 | 22.53 | 0.841 | 0.129 |

Table 15: Performance of our S-NeRF by using 3D bounding boxes in different qualities (mIoU).

| Method | Training time | Inference time (resolution) | PSNR |
|---|---|---|---|
| MipNeRF | 17 hours | 370s | 17.34 |
| MipNeRF-360 | 12 hours | 210s | 23.17 |
| Ours | 17 hours | 370s | 25.68 |
| Ours with distill mode of MipNeRF-360 | 12 hours | 210s | 25.35 |

Table 16: Evaluation of the training time and the inference speed (on an RTX A6000 GPU).

| Method | PSNR | SSIM | LPIPS |
|---|---|---|---|
| BARF Lin et al. (2021) | 22.23 | 0.751 | 0.451 |
| BARF w/o initialization | 12.16 | 0.527 | 0.634 |
| SCNeRF Jeong et al. (2021) | 25.15 | 0.784 | 0.377 |
| SCNeRF w/o initialization | 14.24 | 0.578 | 0.583 |
| Ours | **25.68** | **0.788** | **0.375** |

Table 17: Comparisions of NeRF–, BARF and SCNeRF in training our S-NeRF

| Camera | PSNR | SSIM | LPIPS |
|---|---|---|---|
| Front/Back views | 22.22 | 0.731 | 0.389 |
| Left/Right views | 21.17 | 0.681 | 0.498 |

Table 18: Rendering qualities of the front views and side views.

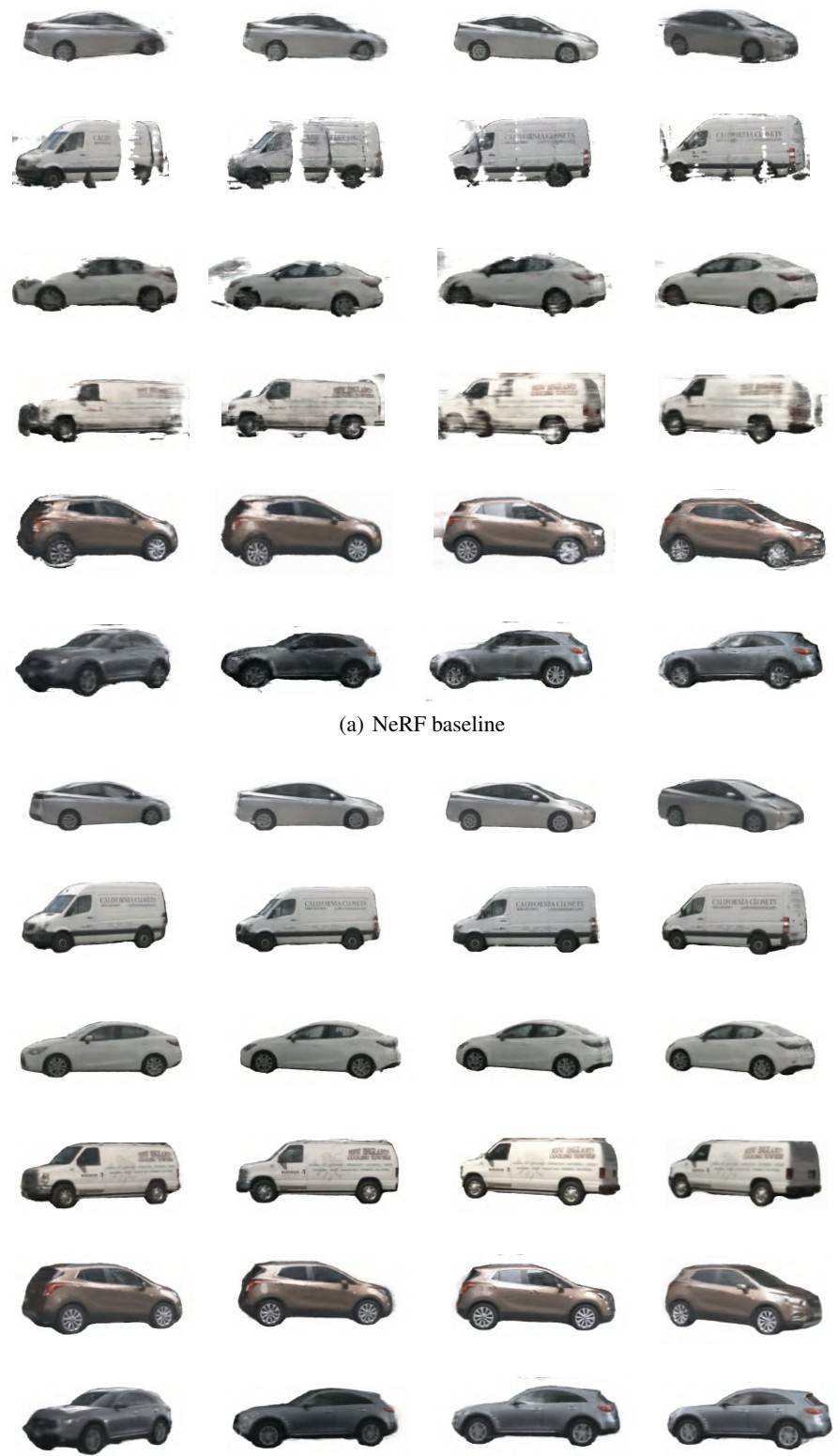

(a) NeRF baseline

(b) Our S-NeRF

Figure 14: Comparisons with NeRF baseline for foreground car rendering. Four different novel views are rendered for five different cars. Our S-NeRF significantly reduce the "floats", blurs and other artifacts.

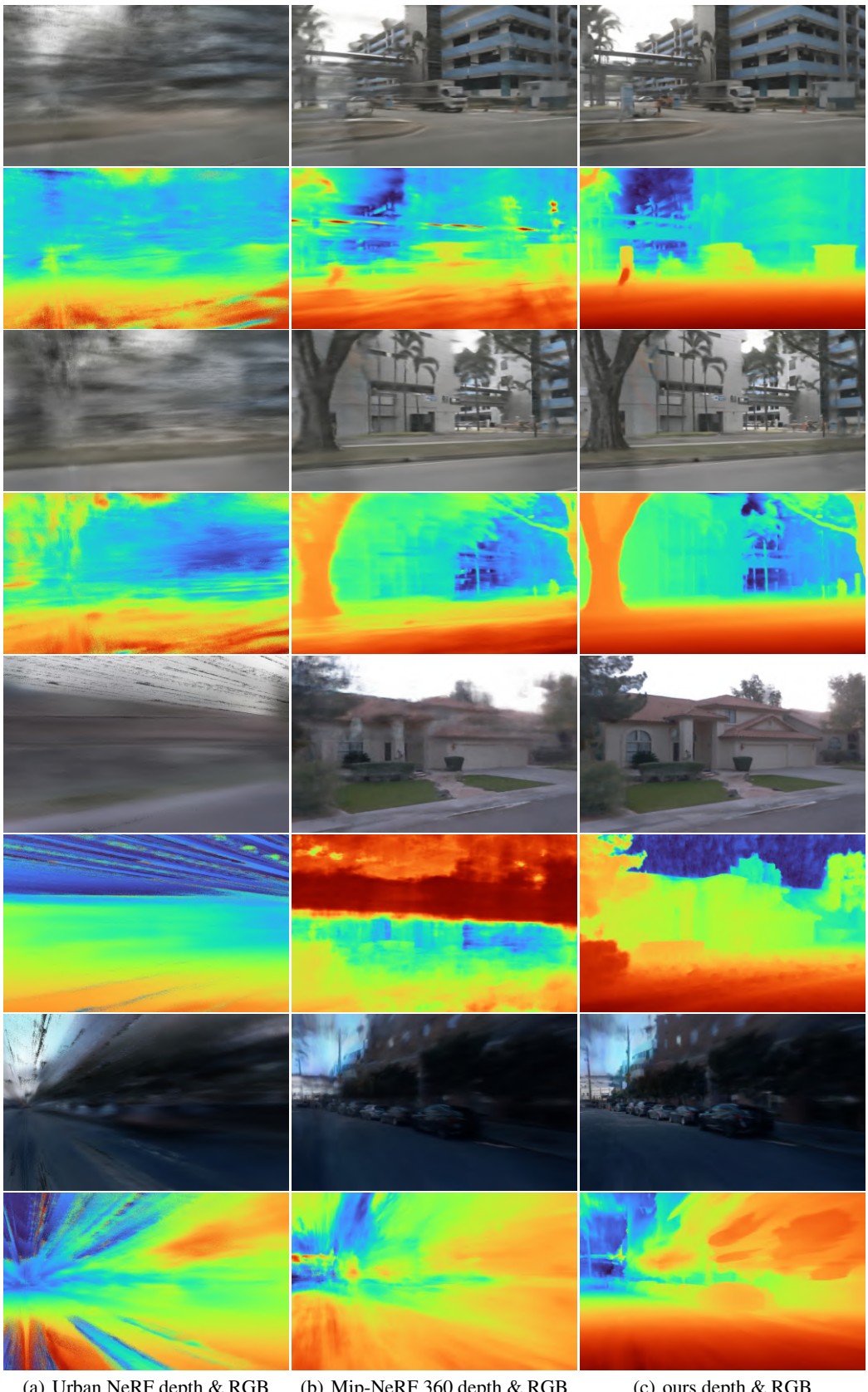

(a) Urban NeRF depth & RGB     (b) Mip-NeRF 360 depth & RGB     (c) ours depth & RGB

Figure 15: Comparisons with state-of-the-art Mip-NeRF 360 Barron et al. (2022) and Urban-NeRF Rematas et al. (2022).

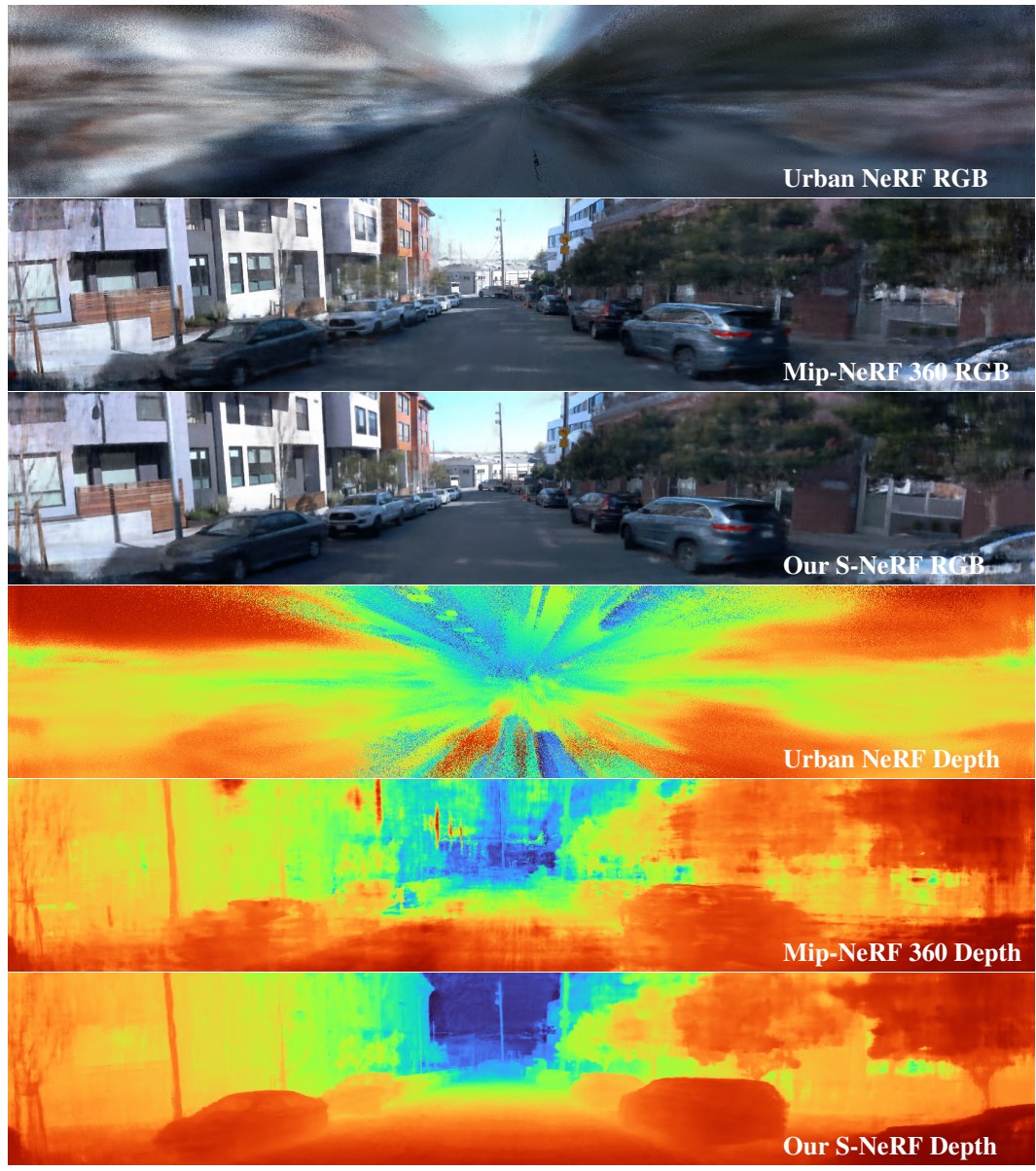

Figure 16: We render more 180 degree panorama views for visual comparisons. Scenes are from the Waymo datasets.

