# OpenReview forum: "S-NeRF: Neural Radiance Fields for Street Views"
_ICLR.cc/2023/Conference — ICLR 2023 poster_

### Official Review · Reviewer_TJeu · 2022-10-24

**Confidence:** 3
**Correctness:** 2
**Technical Novelty And Significance:** 2
**Empirical Novelty And Significance:** 2
**Recommendation:** 6

**Clarity, Quality, Novelty And Reproducibility:**

Building on existing approaches, the framework proposed in this paper addresses a research question with an efficient approach. This paper draws on so many existing techniques such as all kinds of NeRF, 3-D object detection and common depth estimation, that it's more like a high-level combination. So I think the innovative newness of this paper still needs to be strengthened and can be followed up with more in-depth work.

**Strength And Weaknesses:**

+ Strengths：
● The paper is well-written and easy to follow.
● The depth supervision design is novel and technically sound. The ablation study verifies its contribution as well. More importantly, such a strategy allows us to use low-cost spare Lidar data instead of the dense one in urban-nerf.
● The experiments show the proposed method outperforms SOTA works in both foreground vehicles and background scenes NVS.

- Weaknesses：
● The virtual digital camera transformation process appreciably depends on the quality of 3D detection results. This possible threat should lead to an inconsistent coordinate system transformation.
● Ablation study:  Based on the ablation study in tab 3 and fig 6, the depth confidence surely helps the proposed model. However, the mechanism behind L_depth is still blurry. It is valuable and theoretically interesting to conduct a more detailed ablation study on confidences of rgb, ssim, vgg, depth and flow and show what technology is critical to this work.
● Discussion about deeper principles: It is not clear why the proposed method gives significantly better results than other methods for background composition using only RGB images. We also need more analysis on the limitations of the proposed method. It would be good if the authors provide some failure case  evaluation to better understand the limitation of the proposed method.
● Make the streets more real: Dynamic objects such as pedestrians, bicycles, and trucks are very common on the streets. Although it is too demanding to reconstruct dynamic objects, it is interesting to study the effect of these dynamic objects on the reconstruction of the target.

**Summary Of The Paper:**

This paper addresses the novel view synthesis of street view and contributes a new NeRF design (named S-NeRF). Since the conventional nerf solutions could easily fail in street view scenes, the authors use a new scene parameterization function and camera poses for learning better neural representations from street views. The experiments demonstrate the proposed method outperforms state-of-the-art works on general self-driving datasets.

**Summary Of The Review:**

Overall, this paper presents a novel Nerf framework for self-driving applications. But I still think the author could have done more in-depth research and tweaking.

---

> ### Author Response · Authors · 2022-11-17
> **To reviewer TJeu**
>
> **1) Virtual camera transformation depends on 3D detection results:**
>
> *We already provided detailed comparisons and analysis in Appendix C.4 "Influence of quality of 3D detection results".*
> Please refer to Table 12 of the appendix. When the 3D detector fails to detect accurate 3D bounding boxes for the foreground objects, our S-NeRF still produces good results with a just slight drop in the rendering quality (E.g. 0.49 $\downarrow$in PSNR when the mIoU is 0.67 and 0.97 $\downarrow$for a poor mIoU of 0.55). This is because we also implement pose refinement (Sec. 3.2 of the paper), the multi-view photometric and depth consistency will assist the pose refinement/correction during the training of our S-NeRF. The accuracy of the 3D detection results would not significantly influence our final results on foreground objects. Even using the poor detection results, our S-NeRF (22.53 in PSNR) still performs far better than the Geosim [3] (15.12 in PSNR). We observed that even the poorly detected bounding boxes ( around 0.5 in IoU) are effective for training our S-NeRF. These results (>0.5 in IoU) can be easily realized by many existing 3D object detectors.
>
> **2) Depth supervision, a more detailed ablation study on confidences.**
>
>  *Referring to Table 7 of the Appendix. We already provided detailed comparisons and analysis of the confidence components.*
> We also added new illustration figures to the Appendix to show the mechanism of the depth confidences.
> We find geometry confidences (depth consistency and flow consistency) play more important roles in reducing depth errors.
>
> **3) Why S-NeRF is significantly better than other NeRFs using only RGB images.**
>
> It's difficult to train existing NeRFs (e.g. Mip-NeRF 360, Mip-NeRF) on the self-driving dataset (e.g. waymo, nuScenes) for driving simulation due to:
>
> - Fewer views. The ego car moves fast and some objects or contents only appeared in two or three images.
> - Multi-view photometric inconsistency: illumination variations, noises, reflections, and large smooth regions in the driving images.
> - Difficult pose estimation. COLMAP and other SFM methods fail in these multi-camera driving scenes.
>
> All these make it difficult to build accurate 3D geometry by using only photometric consistency in state-of-the-art NeRFs (e.g Mip-NeRF, Mip-NeRF 360). Our method utilizes confidence-guided depth supervision that helps to create accurate 3D geometry and refine the inaccurate camera poses for better volume rendering.
>
> **4) Failure case analysis:**
>
> *We already discussed the failure cases and provided a detailed analysis in Table 15 and Sec. D "Failure Cases" of the appendix.*
> Sometimes, when the ego car (for data collection) moves fast, fewer images are captured by left or right cameras, then it will produce blur rendering for left and right views.
>
> **5）Other dynamic objects:**
>
> Please refer to Fig. 4~6 in the Appendix. Similar to the moving car, we reconstruct a moving truck using only 4 image views. The quality of the novel-view rendering (Fig. 4) is good enough for our driving simulation. For the dynamic persons, since only a limited number of image views can be captured for a single person in the nuScenes and Waymo datasets, and the person is also walking with varying poses, it is difficult to reconstruct high-quality 3D person using only a few (2-5) views. We instead use existing monocular video data to reconstruct 3D persons and render novel views with novel poses. Fig. 5 shows an example using Anim-NeRF [1] to reconstruct persons in the People-Snapshot dataset [2]. Then, we can combine the 3D persons (with novel views and novel poses) with the S-NeRF scenes for the realistic driving simulation (Fig. 6). Similarly, other objects (e.g. bicycles) can be first reconstructed and then combined into our S-NeRF scenes for driving simulation.
>
> ---
>
> [1] Animatable Neural Radiance Fields from Monocular RGB Videos, Jianchuan Chen et al, Arxiv 2021.
> [2] Video Based Reconstruction of 3D People Models, CVPR 2018.
> [3] GeoSim: Realistic Video Simulation via Geometry-Aware Composition for Self-Driving, CVPR 2021.

---

### Official Review · Reviewer_SFqp · 2022-10-25

**Confidence:** 3
**Correctness:** 4
**Technical Novelty And Significance:** 3
**Empirical Novelty And Significance:** 3
**Recommendation:** 6

**Clarity, Quality, Novelty And Reproducibility:**

The wring of this paper is good and easy to follow. This paper clearly states which parts are from other papers and which parts are proposed by themselves. The originality of the work is good.

**Strength And Weaknesses:**

Strength:

+ Although the idea of using sparse and noisy Lidar depth as supervision is not new in the community, the proposed method of using confidence scores and the learnable weights to balance the different items to generate confidence scores is interesting and smart.

+ Experimental results seem to be very promising.

Weakness:

The overall contribution of this paper is a little insignificant. For example,
(1) one of the claimed contributions, " improve the scene parameterization", is just a small modification on the scene coordinates representation strategy, shown in Eq. (2);
(2) The "improved camera poses" illustrated in Sec 3.3 are just some engineering processing tricks.

One of the unique challenges in the street-view neural radiance field is that it contains static background and dynamic objects (vehicles, pedestrians). I was expecting a novel neural radiance field method that could distinguish the two kinds of scenes automatically. However, this paper adopts an existing method to first detect the dynamic object (vehicle) and estimates its pose, and then apply a NeRF-type method to generate the neural radiance field. This is less attractive.

Other comments:
(1) The paper below also addresses the neural radiance field of unbounded scenes. It may be worth a discussion in the related work.

Li et al. Neural plenoptic sampling: capturing light field from imaginary eyes. 2021

(2) The second row in Sec 4.5, "out"--> "our"?

**Summary Of The Paper:**

This paper addresses the problem of neural radiance fields for street views. Considering the low overlap among street view images and the unbounded nature of street scenes, this paper improves the scene coordinate parameterization of existing work and proposes a new method for using sparse and noisy lidar depth as supervision. Experimental results show the performance of the improved method is significantly superior to previous methods.

**Summary Of The Review:**

Although I have concerns about the insignificance of some of the contributions, the idea of how to use sparse and noisy lidar points as supervision is interesting and should benefit related tasks where sparse and noisy lidar points are available.

---

> ### Author Response · Authors · 2022-11-17
> **To reviewer SFqp**
>
>
> **1) The separation of the static background and dynamic objects and contribution:**
>
> _Note this is not exactly the same to the problem of distinguishing the static background and dynamic objects in the dynamic photography_.
> We aim at driving simulation, in which the dynamic objects should be fully controllable and move with controlled speeds and trajectories. Therefore, the background scenes and the foreground cars have to be reconstructed separately.
> This is impracticable for the existing dynamic NeRFs [2][3], in which the rotation, speed, location, and trajectory of the object cannot be modified.
> In addition, the driving simulation also requires highly accurate 3D depth information and 3D bounding box labels for object placement and occlusion handling which are difficult for the existing alternatives (e.g. MipNeRF, MipNeRF-360).
>
> In this work, for the first time, we propose to train NeRF reconstruction on the challenging driving dataset (collected by self-driving cars) for driving simulation.
> Specifically, we are able to render the objects and background scene to simulate various driving situations (including unseen cases in the real world). We have provided more qualitative results in our revision.
> Compared with our separated learning, we find that learning a single model for both the background and the moving objects produces poor results for dynamic things due to geometry inconsistency and the limited number of image views (only 2-4 views are available for many moving objects).
>
> Extensive experiments on the large-scale driving datasets (e.g., nuScenes and Waymo) demonstrate that our method beats the state-of-the-art rivals by reducing 7∼ 40% of the mean-squared error in the street-view synthesis and a 45% PSNR gain for the moving vehicles rendering.
> **We strongly advocate that, in a nutshell, our approach per se significantly revolutionizes the learning-based driving simulation.**
>
> **2) New Reference.**
>
> We have added [1] to the related work for comparisons and analysis. It proposes to use a Multi-Layer Perceptron (MLP) as an approximator to learn the plenoptic function and represent the light-field in NeRF. We believe this can also be used in our S-NeRF to improve the rendering of lighting variations.
>
> **3) Typos:**
>
> Many thanks. We have made a revision to make it even clear.
>
> ---
>
> [1] Neural plenoptic sampling: capturing light field from imaginary eyes. 2021
>
> [2] D-NeRF: Neural Radiance Fields for Dynamic Scenes, CVPR 2021.
>
> [3] Neural Scene Flow Fields for Space-Time View Synthesis of Dynamic Scenes, CVPR 2021.

---

### Official Review · Reviewer_k6S3 · 2022-10-30

**Confidence:** 4
**Correctness:** 4
**Technical Novelty And Significance:** 4
**Empirical Novelty And Significance:** 4
**Recommendation:** 8

**Clarity, Quality, Novelty And Reproducibility:**

The paper is well written, although the flow of the paper could be improved. The different blocks of their system pipeline are explained but it is challenging to relate the different blocks. This makes reading the paper a bit difficult requiring going back and forth to understand the blocks. Also, the number of figures is very limited to understand the various equations. The authors should have put more effort in putting more figures to geometrically explain their error functions. The paper also uses a number of existing deep learning work e.g. densifying sparse point cloud, camera pose improvement to name a few as their building blocks. Their is no clarity of explanation on what happens when these blocks do not perform well. From the results, it appears that the results are better than existing methods for street view scenes. This may imply that when the system works, it works very good but would have helped to have some analysis of cases where it failed and why. So, reproducibility may be a challenging factor with this paper.

**Strength And Weaknesses:**

Strength:
* The results are good and show that their method works well.

Weakness:
* Sec 3.3: One can use the 3D....the ego car. How is the relative calibration computed. Some more details would be good for clarity.
* Eq.4 is not correct. The correct equation should be (Pi * Pb^-1)^-1 = Pb * Pi^-1
* Sec 3.4 requires some figures and images to clearly explain the different confidence measures and how they are computed.
* In "LiDAR depth completion", when the LIDAR depths are accumulated, they are from different time instants. So, how are they combined and brought to the same coordinate system. For static scenes, the extrinsics between the car can be used to fuse the depths, but for moving objects in the scene like cars this will not work and they will always end up as outlier depths. This whole paragraph is not clearly explained as it appears to be one of the contributions of this paper.

**Summary Of The Paper:**

The paper proposes a method for novel view rendering using deep learning based on NERF model where the background and the foreground are synthesized jointly. The NERF model is given extra supervision via LIDAR depths and different confidence maps based on LIDAR depth. The confidence maps help in incorporating standard street view datasets which come with noisy LIDAR depths. A number of existing work is used as building blocks in the paper.

**Summary Of The Review:**

In summary, the paper has some novel contributions in the way they incorporate LIDAR data and confidence measures which enable use of existing standard datasets for training. The writing could have been improved as the flow of the paper is not smooth. More figures should have been put for better explanation of the different blocks.

---

> ### Author Response · Authors · 2022-11-17
> **To reviewer k6S3**
>
> **1) The computation of relative calibration (Sec 3.3):**
>
> We provided a new illustration figure (Fig. 4) and improve the contents of this section to make it more clear.
> Referring to Fig. 4 of the revised paper, during the data collection, the ego car (cameras) is moving and the target car (object) is also moving. To train NeRF reconstruction, the virtual camera representation transforms the "dynamic" scenes into a static setting, i.e. it treats the target car (moving object) as static and then computes the relative camera poses for the ego car's cameras. These relative camera poses can be estimated through the 3D object detectors. After the transformation, only camera is moving which is favorable in training NeRFs.
>
> **2) Eq.4 is not correct**
>
> Thanks. Eq. (4) (Eq. 2 in the revised paper) is updated.
>
> **3) Confidence figures and illustrations**
>
> We visualized and compared the confidence maps in (Appendix Fig. 2) of the Appendix. The computation of the geometry and projection confidence is also illustrated in (Appendix Fig. 3). We also provide an overview illustration figure (Appendix Fig. 1) of the S-NeRF framework.
>
> **4) Accumulate the LiDAR points, remove outliers of moving objects.**
>
> Given the LiDAR positions of the consecutive frames $[P_{t-1}, P_t, P_{t+1}]$ at different time $[t-1, t, t+1]$, we are able to project the 3D LiDAR points at time t-1 and t+1 to t by $\Delta P$ between $P_t$ and $P_{t-1}/P_{t+1}.$ This is similar to the mapping function $\psi$ in Eq. (4) of the paper. Most outliers of the moving objects and other regions can be removed by the consistency check using the optical flow and the depth projection (Eq. (6-7) of the paper). The rest outliers can be handled by the proposed confidence-guided learning. Please see the section B.1 of appendix for more details.
>
> **5) Improve the writing.**
>
> We have improved the writing and provided more illustration figures to the paper and appendix to make it easy to understand.
>
> **6) No clarity of explanation on what happens when these blocks do not perform well.**
>
> In the appendix, we already provided experiments and analysis of the effects of different blocks/components. Here we give a summary of them:
>
> - **Depth quality:** Please refer to Table 1 of the rebuttal to Reviewer u7q4 and Table 10 (Appendix Sec. C.4 "Depth quality") of the Appendix. Using worse depth completion or poor depth maps doesn't significantly influence the final rendering quality.
> - **Confidence measure:** Please refer to Table 7 of the Appendix: "Ablation study on confidence components". Dropping some of the confidence component, the rendering quality (PSNR) drops by 0.02-0.22%, and the depth error increases by 0.07-2.42%. Discard all the confidences, the PSNR drops by 0.9% and the depth error rate increases by 6.27%. We also find that the geometry confidence is more important in reducing depth errors.
> - **3D detector for foreground objects:** Please refer to Table 12 of the Appendix. When the 3D detector fails to detect accurate 3D bounding boxes for the foreground objects, our S-NeRF still produces good results with a slight drop in the rendering quality (E.g. 0.49 $\downarrow$when the mIoU is 0.67 and 0.97 $\downarrow$for a poor mIoU of 0.55). Even using the poor detection results, our S-NeRF (22.53 in PSNR) still performs far better than the Geosim [1] (15.12 in PSNR). We observed that even the poorly detected bounding boxes ( around 0.5 in IoU) are effective for training our S-NeRF. These results (>0.5 in IoU) can be easily realized by many existing 3D object detectors.
> - **Pose processing:** Please refer to Table 14 and Sec. C.4 "Different pose refinement" of the Appendix. We find that using the default pose refinement performs the best. SC-NeRF can also be used for pose refinement to produce good rendering results. Besides, pose initialization plays an important role in the pose processing step. Without pose initialization, the rendering quality drops significantly.
>
> **7) No analysis of Failure cases.**
>
> We already discussed the failure cases and provided a detailed analysis. These are presented in Table 15 and Sec. D "Failure Cases" of the appendix.
>
> ---
> [1] GeoSim: Realistic Video Simulation via Geometry-Aware Composition for Self-Driving, CVPR 2021.

---

### Official Review · Reviewer_u7q4 · 2022-10-30

**Confidence:** 2
**Correctness:** 3
**Technical Novelty And Significance:** 1
**Empirical Novelty And Significance:** 2
**Recommendation:** 6

**Clarity, Quality, Novelty And Reproducibility:**

(minor concerns) My impression is that the paper is hard to understand and not polished enough.

- Writing quality seems to drop noticeably starting around section 3.2.
- No system figure, no illustration of the contribution.
- What are we supposed to learn from this paper, what questions does it answer?

In particular, I had a hard time understanding the "virtual camera transformation" part. It seemed related to the separation of the moving objects, but it was not clear to me otherwise.

**Strength And Weaknesses:**

Strengths

- Achieving robustness to noisy depth in this setting is a practical goal.
- Using depth completion priors and multi-view reprojection is a reasonable approach.

Weaknesses

- I suspect the improvement is due to the separate evaluation of static background and moving foreground objects (how was this done for the baselines?), on top of the heavy usage of the pre-trained NLSPN depth completion network.

**Summary Of The Paper:**

This paper (S-NeRF) aims to train a NeRF using images from driving datasets (e.g., nuScenes and Waymo) with a limited number of views (2-6 images). The improvement over Urban-NeRF is that it densifies sparse LiDAR depth for depth supervision, through a reprojection-based confidence map approach. The main claimed advantage is robustness to noise in the depth point cloud. The experiments show sharper, more recognizable novel view synthesis output, over Urban-NeRF and Mip-NeRF.

**Summary Of The Review:**

The paper lacks technical novelty, despite the increased complexity (see weaknesses). If I understand correctly, the main contribution of the paper is showing that novel view synthesis of static and dynamic objects can benefit from depth densification and multi-camera reprojection. In my view, the impact of this finding alone is below threshold for publication at ICLR.

---

> ### Author Response · Authors · 2022-11-17
> **To reviewer u7q4 [Part 2/2]**
>
> **4) Improvements come from separation evaluation:**
>
> No. All evaluations are conducted under same setting. The improvements mainly come from the proposed confidence-guided depth supervision, better scene parameterization and pose processing.
> In Table 2 (Section 4.3), all methods are evaluated under the same setting. Since state-of-the-art alternatives (MipNeRF/MipNeRF-360, Urban-NeRF) cannot handle moving objects, whilst we only use static backgrounds (including static objects, like cars) in training and evaluation for fair comparisons in Table 2.
> In Table 1 (Section 4.2), we evaluate the performance of reconstructing the foreground cars. We compare our S-NeRF with the standard object-centric NeRF and the driving simulation method Geosim [1]. Since the object-centric NeRF can not be used to reconstruct the background scenes, in this experiment, only the foreground cars are used for evaluation.
>
> **5) Heavy usage of the pre-trained NLSPN.**
>
> Our S-NeRF does not heavily rely on the depth completion network (NLSPN). We already provided experiments and analysis in Appendix B.4 “Depth quality” (Table 9). It shows that the depth quality after depth completion does not significantly influence the rendering quality. When depth errors increase with poor depth maps, the PSNR of the novel views just slightly dropped by 0.03-0.12%.
> We also test another two worse depth completion methods by replacing the NLSPN with [2] or [3]. Even using the worse traditional depth completion method [2] (without learning a deep neural network), our S-NeRF still achieves a similar rendering quality (24.41->24.40). This experiment shows that our method does not rely on NLSPN depth quality. Benefiting from our confidence-guided depth supervision, many other depth completion methods can be also used in our S-NeRF.
>
> Table 1: Effects of different depth completion methods.
>
> | Methods | Performance on KITTI depth complete task (Mean absolute error[mm]) | Rank on KITTI leaderboard | PSNR | SSIM | LPIPS |
> | :-- | :-- | :-- | :--: | :--: | :--: |
> | Traditional depth completion [2] | 302.60 | 116 | 24.40 | 0.782 | 0.344 |
> | Sparse Depth Completion [3] | 215.02 | 61 | 24.50 | 0.785 | 0.344 |
> | Our default settings (NLSPN [4]) | 199.59 | 40 | 24.41 | 0.783 | 0.345 |
>
> **6) Writing quality drops around section 3.2**
>
> We have improved the presentation and added new illustration figures to make it even clear.
>
> **7) System figure, illustration of the contribution.**
>
> We added a system figure to the Appendix (Fig. 1, considering the space limit in the main paper).
>
> **8) Virtual camera transformation**
>
> We provided a new schematic illustration (Fig. 4 of the paper) and improved the contents of this section to make it more clear.
> Referring to Fig. 4, during the data collection, the ego car (cameras) is moving and the target car (object) is also moving. To train NeRF reconstruction, the virtual camera representation transforms the "dynamic" scenes into a static setting. It transforms the target car (moving object) as static one and then computes the relative camera poses for the ego car's cameras. These relative camera poses can be estimated through the 3D object detectors. After the transformation, only the camera is moving which is favorable in training NeRFs.
>
>
> ---
>
> [1] GeoSim: Realistic Video Simulation via Geometry-Aware Composition for Self-Driving, CVPR 2021.
> [2] In Defense of Classical Image Processing: Fast Depth Completion on the CPU, CVPR 2018.
> [3] Sparse and noisy LiDAR completion with RGB guidance and uncertainty, CVPR 2019.
> [4] Non-Local Spatial Propagation Network for Depth Completion, ECCV 2020.

---

> ### Author Response · Authors · 2022-11-18
> **To Reviewer u7q4 [Part 1/2]**
>
> **1) Motivation**
>
> **(i)** Driving simulation is critical for the highly functional autonomous driving system in aspect of perception, motion prediction/planning and even control. However, existing graphic-modeling-based unreal driving simulators (e.g. CARLA, AIRSIM) are ineffective to generate useful data for the real-world self-driving.
>
> **(ii)** On the other hand, neural radiance fields (NeRFs) achieves great success on synthesising novel views of objects and scenes, given the object-centric camera views with large overlaps.
> However, we conjugate that this paradigm does not fit the nature of the street views that are collected without much overlapping by the self-driving car in large-scale unbounded scenes.
> It is also impracticable to reconstruct the moving vehicles with existing NeRFs, in which the rotation, speed, location, and trajectory of the object cannot be modified.
>
> **(iii)** The motivation behind our work is that we need to solve above dilemma and open the door for the learning-based approach to render various large-scale long-range background scenes and moving vehicles on the challenging driving dataset.
> **We strongly advocate that, in a nutshell, our approach per se significantly revolutionizes the learning-based driving simulation.**
>
>
> **2) Challenges**
> It is hard to train a NeRF model on the self-driving datasets (e.g. waymo, nuScenes) for driving simulation due to:
>
> **(i)** Difficult pose estimation. COLMAP and other SfM methods fail in these multi-camera driving scenes (both for foreground objects and background scenes).
>
> **(ii)** Large-scale unbounded scenes. A nuScenes/Waymo sequence usually present a scene with range more than 200m.
>
> **(iii)** Fewer views. A typical Waymo/nuScenes sequence covers 200m range but contains only 100-300 images. The ego car moves fast and some objects or contents only appeared in two or three images.
>
> **3) Contribution**
>
> **(i)** For the first time, we propose a learning-based approach characterized by the formulation of Neural Radiance Fields, dubbed S-NeRF, for driving simulation.
>
> **(ii)** This is achieved by innovatively learning with confidence guided LiDAR depth supervision and improving the scene parameterization function and the camera poses for learning better neural representations of both large-scale long-range (e.g. 200m) background scenes and moving vehicles on the challenging driving dataset per se.
>
> **(iii)** Extensive experiments on the large-scale driving datasets (e.g., nuScenes and Waymo) demonstrate that our method beats the state-of-the-art rivals by reducing 7∼ 40% of the mean-squared error in the street-view synthesis and a 45% PSNR gain for the moving vehicles rendering.

---

> ### Author Response · Authors · 2022-12-05
> **Request for feedback on the rebuttal**
>
> Dear Reviewer u7q4,
>
> We appreciate all the reviewing time and effort. With our best appreciation, we have made the revised paper, supplementary materials and the response in detail to each individual comment. While we consider this could have addressed all the concerns raised hopefully, it is most critical that the reviewer can kindly read our response and tell us how the issues have been addressed and if any concerns are left to be addressed. We would take all the comments/suggestions as carefully as possible and address them with our best efforts. Many thanks for every effort the reviewer made and will make on our work.
>
> Best wishes,
> Authors

---

### Author Response · Authors · 2022-11-17
**General response to all reviewers**

Thanks for the valuable comments. We have revised our paper and the revisions are highlighted in red. Here we briefly summarise our motivation and contributions.

**Motivation**

**(i)** Driving simulation is critical for the highly functional autonomous driving system in aspect of perception, motion prediction/planning and even control. However, existing graphic-modeling-based unreal driving simulators (e.g. CARLA, AIRSIM) are ineffective to generate useful data for the real-world self-driving.

**(ii)** On the other hand, neural radiance fields (NeRFs) achieves great success on synthesising novel views of objects and scenes, given the object-centric camera views with large overlaps.
However, we conjugate that this paradigm does not fit the nature of the street views that are collected without much overlapping by the self-driving car in large-scale unbounded scenes.
It is also impracticable to reconstruct the moving vehicles with existing NeRFs, in which the rotation, speed, location, and trajectory of the object cannot be modified.

**(iii)** The motivation behind our work is that we need to solve above dilemma and open the door for the learning-based approach to render various large-scale long-range background scenes and moving vehicles on the challenging driving dataset.
**We strongly advocate that, in a nutshell, our approach per se significantly revolutionizes the learning-based driving simulation.**


**Challenges**
It is hard to train a NeRF model on the self-driving datasets (e.g. waymo, nuScenes) for driving simulation due to:

**(i)** Difficult pose estimation. COLMAP and other SfM methods fail in these multi-camera driving scenes (both for foreground objects and background scenes).

**(ii)** Large-scale unbounded scenes. A nuScenes/Waymo sequence usually present a scene with range more than 200m.

**(iii)** Fewer views. A typical Waymo/nuScenes sequence covers 200m range but contains only 100-300 images. The ego car moves fast and some objects or contents only appeared in two or three images.

**Contribution**

**(i)** For the first time, we propose a learning-based approach characterized by the formulation of Neural Radiance Fields, dubbed S-NeRF, for driving simulation.

**(ii)** This is achieved by innovatively learning with confidence guided LiDAR depth supervision and improving the scene parameterization function and the camera poses for learning better neural representations of both large-scale long-range (e.g. 200m) background scenes and moving vehicles on the challenging driving dataset per se.

**(iii)** Extensive experiments on the large-scale driving datasets (e.g., nuScenes and Waymo) demonstrate that our method beats the state-of-the-art rivals by reducing 7∼ 40% of the mean-squared error in the street-view synthesis and a 45% PSNR gain for the moving vehicles rendering.


---
[1] GeoSim: Realistic Video Simulation via Geometry-Aware Composition for Self-Driving, CVPR 2021.

---

### Decision · Program_Chairs · 2023-01-20

**Decision:**

Accept: poster

**Justification For Why Not Higher Score:**

The approach here is a combination of multiple tricks and may not offer any interesting insights/techniques that maybe useful generally beyond the specific setting studied.

**Justification For Why Not Lower Score:**

This work could be is empirically solid and builds a useful system that outperforms prior neural 3D reconstruction approaches in outdoor driving scenes. After the discussion phase, all reviewers recommend acceptance.

**Metareview: Summary, Strengths And Weaknesses:**

The reviewers raised concerns regarding the technical contributions e.g. that the proposed approach significantly builds on prior work with minor modifications, and side-steps the key challenges of dealing with dynamic objects using off-the-shelf (supervised) 6D pose estimation.  However, on the positive side, the reviewers appreciate the setting studied and the impressive results in the outdoor driving setup. The paper also makes some interesting contributions in how to robustly incorporate LiDAR observations for training NeRFs and presents a useful system with good results in reconstructing neural fields in outdoor driving scenarios. After internal discussion, the reviewers all leaned towards acceptance as this work and the AC concurs.

**Note From Pc:**

if the above contains the word "oral" or "spotlight" please see: "oral" presentation means -> notable-top-5% and "spotlight" means -> notable-top-25%. As stated in our emails, we are disassociating presentation type from AC recommendations

**Summary Of Ac-Reviewer Meeting:**

Unfortunately, I was not able to have a meeting for this work due to travel constraints, although I tried initiating discussion on OpenReview. I think the reviewers agree on the strengths and weaknesses of the work, and the ratings converged to all accepts after the online discussion.